# Specific PIP$_2$ binding promotes calcium activation of TMEM16A chloride channels

Zhiguang Jia[1,2] & Jianhan Chen [1,2✉]

TMEM16A is a widely expressed $Ca^{2+}$-activated $Cl^-$ channel that regulates crucial physiological functions including fluid secretion, neuronal excitability, and smooth muscle contraction. There is a critical need to understand the molecular mechanisms of TMEM16A gating and regulation. However, high-resolution TMEM16A structures have failed to reveal an activated state with an unobstructed permeation pathway even with saturating $Ca^{2+}$. This has been attributed to the requirement of PIP$_2$ for preventing TMEM16A desensitization. Here, atomistic simulations show that specific binding of PIP$_2$ to TMEM16A can lead to spontaneous opening of the permeation pathway in the $Ca^{2+}$-bound state. The predicted activated state is highly consistent with a wide range of mutagenesis and functional data. It yields a maximal $Cl^-$ conductance of ~1 pS, similar to experimental estimates, and recapitulates the selectivity of larger $SCN^-$ over $Cl^-$. The resulting molecular mechanism of activation provides a basis for understanding the interplay of multiple signals in controlling TMEM16A channel function.

[1] Department of Chemistry, University of Massachusetts, Amherst, MA, USA. [2] Department of Biochemistry and Molecular Biology, University of Massachusetts, Amherst, MA, USA. ✉email: jianhanc@umass.edu

Ca[2+]-activated Cl[−] channels (CaCCs) direct the flow of information from calcium signaling to Cl[−] current and are involved in numerous physiological functions including neuronal excitation, smooth muscle contraction, and airway fluid secretion[1–5]. After two decades of searching, the molecular identity of CaCC was finally determined as TMEM16A[2,6–8]. TMEM16A, also known as anoctamin-1 (ANO1), belongs to the TMEM16 family of multi-functional proteins that include both CaCC and Ca[2+]-activated lipid scramblase[5]. Two members of this family, TMEM16A and B, are CaCCs, while TMEM16C, D, E, F, G, J, and K are phospholipid scramblases[5]. Some of TMEM16 scramblases also possess nonselective ion channel activities[9]. TMEM16A, in particular, has been shown to play key roles in vital functions such as mucin secretion and airway surface liquid homeostasis[10,11], chloride secretion in salivary gland[12], and airway smooth muscle contraction[10]. TMEM16A has also been also found to be overexpressed in many cancers and is considered a promising anti-cancer drug target[3]. Despite these important roles, the molecular mechanisms of TMEM16A CaCC activation and regulation are only starting to be understood[3,13].

Functional TMEM16A channels are double-barreled homodimers with each monomer harboring a conduction pore that can be activated independently[14,15]. High-resolution structures of TMEM16A have been determined in both the Ca[2+]-bound and free states using cryo-EM[16–18]. The structures show that each TMEM16A monomer consists of 10 transmembrane helices (TM1-10) (Fig. 1A). The ion conducting pathway is mainly lined by TMs 3 to 7 (Fig. 1B, C). It resembles an hourglass with a wide-open intracellular vestibule (formed by TMs 4-7), a narrow "neck" region (lines by TMs 4-6), and an open extracellular vestibule (formed by TM3-6). Both intracellular and extracellular vestibules are highly hydrophilic and well hydrated[18]. The narrowest region of the neck contains two conserved hydrophobic residues, L547 on TM4 and I641 in TM6 (see Fig. 1B). These conserved residues have been shown to be part of an "inner gate" that is likely conserved in all TMEM16 family proteins[19]. Comparison of the Ca[2+]-bound and free structures reveals the Ca[2+] activation mechanism of TMEM16A[16,17]. In the Ca[2+]-free state, the lower half of TM6 bends towards TM4 near G644 and is unfolded beyond G656 (Fig. S1). In the saturated Ca[2+]-bound state, a group of glutamic acid and glutamine residues on TMs 6-8 coordinate with two Ca[2+] ions. These interactions extend the TM6 helix and pull it away from TM4 and towards the TMs 7 and 8 (see Fig. S1). The hinge motion moves the position of G656 by ~15 Å to widen the lower pore region. Similar Ca[2+]-binding-induced conformation transitions have been observed in TMEM16 lipid scramblases such as nhTMEM16[20], TMEM16F[21–23], and TMEM16K[23] (e.g., see Fig. S1D). Curiously, minimal differences have been observed in the rest of the protein. In particular, the neck and upper regions of the pore are similar, such that the neck region of the pore in the Ca[2+]-bound state remains too narrow to support the permeation of anions. The minimal diameter of the pathway is only ~2.0–2.5 Å, well below that of 3.6 Å for hydrated Cl[−] [16]. As such, further dilation of the pore is believed to be necessary to fully activate the channel.

Besides Ca[2+] signaling and membrane potential, the channel activity of TMEM16A is also regulated by phosphatidylinositol (4,5)-bisphosphate (PIP$_2$)[24–28]. Upon prolonged Ca[2+] activation, TMEM16A CaCCs undergo time-dependent current decay even under saturating Ca[2+] concentrations (>1 μM)[26,29]. Such desensitization or rundown after activation can be inhibited and reversed by binding of PIP$_2$[30]. It is likely that the apparently inactive conformations of Ca[2+]-bound TMEM16A captured in the cryo-EM studies reflect a desensitized state[18,30]. The molecular basis of PIP$_2$ regulation of TMEM16A is only beginning to be understood[24,27,30]. Atomistic simulations using the highly mobile membrane mimetic model (HMMM) combined with glutamine scanning experiments have identified several putative PIP$_2$ binding sites near the cytosolic membrane interface, which were suggested to constitute a putative dynamic regulatory network[24,27]. In contrast, a separate alanine scanning study by Le et al. identified a single cluster of residues, R455, R486, K571, and R579, that form a putative PIP$_2$ binding site near the cytosolic interface of TMs 3–5 (Fig. 1B, C)[30]. This specific PIP$_2$ binding site is close to one of the sites identified in glutamine scanning experiments[24,27]. Atomistic simulations further confirmed that PIP$_2$ could spontaneously bind to this site and makes contacts with the key basic residues in the bound state[30]. The identification of the specific PIP$_2$ binding site suggests that the conducting pore of TMEM16A may be divided into two functional modules, where TMs 6–8 form the Ca[2+] binding activation module and TMs 3–5 form the PIP$_2$-binding regulatory module (Fig. 1D). The rest of TMs (1, 2, 9, and 10) can be considered the supporting and dimerization domain. These studies have now firmly established a crucial role of PIP$_2$ in regulating the desensitization of TMEM16A CaCC. A key question remains is how PIP$_2$ and Ca[2+] binding synergistically activate TMEM16A. Addressing this question will also provide the much-needed insights on the nature of an activated TMEM16A CaCC and how it supports anion permeation.

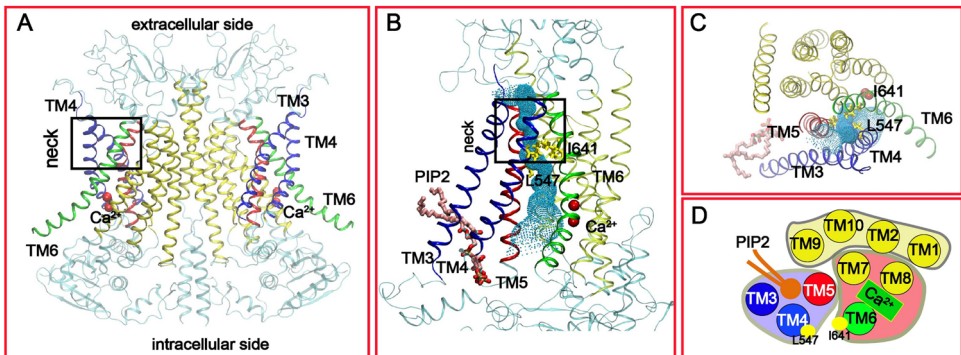

**Fig. 1 Structural features of the Ca[2+]-bound mTMEM16A channel. A** The overall dimer structure with key TM helices highlighted (PDB: 5oyb). **B, C** Front and top views of the putative conducting pore. The pore profile calculated using HOLE[39] is illustrated using the green dots. The two key hydrophobic residues, L547 and I641, in the "neck" region are shown in yellow sticks. The pore-lining TMs are colored in blue (3 and 4), red (5) and green (6), respectively. The other TMs are colored in yellow. The bound PIP$_2$ molecule is represented as sticks in panels **B** and **C**. **D** Cartoon representation of the functional domain organization of TMs from the top. The pore-forming domain consists of the PIP$_2$-binding regulatory module (TMs 3-5) and Ca[2+]-binding activation module (TMs 6-8). TMs 1, 2, 9, and 10 form the dimerization and supporting domain.

In this work, all-atom molecular dynamics (MD) simulations were performed to examine if specific binding of $PIP_2$ to the $PIP_2$-binding regulatory module could promote the activation of $Ca^{2+}$-bound TMEM16A CaCC. Multi-μs simulations in explicit solvent and membrane revealed that specific binding of $PIP_2$ alone was sufficient to induce spontaneous dilation of the pore, increasing the minimal diameter in the neck region from ~2 Å to over 4 Å. The transition mainly involves the movement of TMs 3 and 4 and is transient and reversible, with the dilated state lasting ~1 μs. The pore dilation was not observed in control simulations without $PIP_2$ or in the presence of $PIP_2$ but without $Ca^{2+}$. Conformational clustering analysis allowed the identification of an open state of TMEM16A that contains active ion conduction pathways, with many features consistent with a wide range of existing mutagenesis and functional data. In particular, free energy calculations showed that the predicted open state gave rise to a barrier of ~6 kcal/mol for $Cl^-$ permeation and an estimated maximal conductance of ~1.3 pS. This is similar to the experimental estimates of the single channel maximal $Cl^-$ conductance in the range of below 1 and up to 8 pS[6,8,31–33], suggesting that the predicted open state may correspond to an activated TMEM16A CaCC that has evaded structural studies so far. Free energy analysis further confirmed that the predicted activated state of TMEM16A CaCC would confer selectivity of large $SCN^-$ anions over $Cl^-$, consistent with experimental observations[34,35].

## Results

**Specific $PIP_2$ binding to $Ca^{2+}$-bound TMEM16A induces spontaneous pore opening**. We first performed multi-μs atomistic simulations in explicit solvent and membrane to examine if specific binding of $PIP_2$ to TMEM16A in the $Ca^{2+}$-bound state alone could promote pore opening and channel re-activation. Indeed, we consistently observed spontaneous pore opening events in all three 1.5–3 μs simulations of $Ca^{2+}$-bound TMEM16A in the presence of $PIP_2$ (C16:0/C18:1- $PIP_2$ lipid)) bound to the specific $PIP_2$ binding site in TMs 3–5 previously identified[30] (Fig. 2, Fig. S2 and Supplementary Movie 1, Supplementary Data 2–3). During the simulations, $PIP_2$ maintained stable contacts with the coordinating residues, mainly R455, R486, K571, and R579. The contact probabilities of PIP2 to these four residues are 0.40, 0.49, 0.97, and 0.99, respectively. The average RMSD of the $PIP_2$ headgroup and these basic residues from the initial conformation is $3.5 \pm 0.7$ Å. The minimal distance between TM4 and TM6 at the inner gate increases significantly during the spontaneous opening transitions, which is accompanied by increased hydration of the neck region of the pore (Fig. 2A). In contrast, the neck region is poorly hydrated in the collapsed state (Fig. S3). It can only accommodate at most 1–2 water molecules in a single-file configuration and does not allow ion permeation. Two highly conserved basic residues, K588 and K645 on TM6, appear to play a role in facilitating the hydration of the dilated pore upon $PIP_2$-induced opening (Fig. S3). The charged sidechains of K588 and K645 could rotate and point towards the inner gate region when the hydrophobic inner gate residues, L547 on TM4 and I641 on TM6, become separated during the opening transition. Indeed, both K588 and K645 have been shown to be important for anion permeation[8,36]. The opening transitions appear to be transient and reversible, with the open state lasting ~1 μs. Intriguingly, at most one of the two monomers was observed to adopt the open state at a time in all three trajectories. This is likely due to the stochastic nature of the opening transition and also a consequence of the limited simulation timescales.

In contrast, we did not observe any spontaneous pore opening transition in any of the six simulations of TMEM16A with only $Ca^{2+}$ or $PIP_2$ bound (Fig. 2, Fig. S2 and Supplementary Movies 2 and 3). TMs 4 and 6 remained tightly packed and the pore poorly

hydrated. It should be noted the binding pockets are always occupied with POPC lipids even in simulations without $PIP_2$. These observations are consistent with the experimental results showing that $PIP_2$ is required for maintaining the conductive state of TMEM16A even under saturating $Ca^{2+}$ concentrations[25,26,30]. As additional controls, we further examined the effects of two negatively charged lipids, POPS and PI(4)P, bound to the same pockets. Previous experimental studies have suggested that POPS could not activate TMEM16A, while PI(4)P only shows minor effects on inhibiting channel rundown[25,30]. As summarized in Fig. S2, the binding of these negatively charged lipids only slightly increased the numbers of pore water (*sim 10-15*) compared to those without PIP2 (and with POPC) (*sim 7-9*). Interestingly, an transient pore-opening event was observed in one of the simulations with PI(4)P (*sim 13*), apparently consistent with some capacity of PI(4)P in re-activating the channel.

The $Ca^{2+}$-bound TMEM16A pore conformational space sampled in all six simulations with and without $PIP_2$ (*sim 1-6*) was further characterized using clustering analysis. As the pore opening involves dilating in the whole upper pore region, a single residue–residue distance or the number of pore water molecules alone could not clearly separate the open and closed state. Here, we performed cluster analysis based on distances between pore-lining residues and the number of pore waters in the neck region (See Method part for details). The analysis identified 4 distinct pore conformational states (Fig. S4A). The state representing the open pore conformation (Fig. S4B) is only found in simulations with $PIP_2$. Based on the clustering analysis, two representative structures were extracted from the simulation trajectories to illustrate the conformational difference between the collapsed and open states of the pore (Fig. 2C, D). The predicted open state mainly involves movements in TMs 3-4 and lower pore region of TM5. Specifically, TMs 3 and 4 move away from TMs 5 and 6, increasing the minimal pore radius from ~1 Å to ~2 Å (Fig. 2B and Supplementary Movie 1). The movements of TMs can be further visualized by comparing the distributions of their centers of mass (CMs) in the closed and open states (Fig. S5A). The result shows that there are ~4 and ~1 Å movements of the upper pore segments of TM4 (T539:L547) and TM3 (G510:A523) during activation, respectively, while the other TMs show minimal movements. Note that the structures of all TM helices are very stable as reflected in the small root-mean-squared fluctuation profiles (Fig. S5B). The movement observed in the current simulations is similar to the open–close transition model proposed for other members of TMEM16 family, which involved the reverse movement of TM4 and TM6[37]. In particular, we note that the observed movement of TMs 3–6 is similar to the previously proposed "clam shell" model of TMEM16F activation[19]. However, the degree of TMs movement is smaller in TMEM16A compared to that of TMEM16F lipid scramblase (e.g., see Fig. S1D). The final open pore is still too narrow to accommodate lipid headgroups. Instead, we observed transient insertion of lipid tails in the dilated neck region, for example, around 2 μs mark in *sim 1* (Fig. 2A, blue trace). This may be attributed to the more hydrophobic nature of the upper pore region of TMEM16A, where several hydrophilic residues in TMEM16F (e.g., S514, T606, and T607) are replaced with hydrophobic ones in TMEM16A (e.g., V543, C635, and I636). Interestingly, it has been shown that replacing of hydrophobic residues at the inner gate with charged ones, e.g., L543K, can confer lipid scramblase activity to TMEM16A, presumably by promoting wider dilation of the same ion conducting pore[19,38].

**$PIP_2$-induced open state of TMEM16A is conductive to $Cl^-$.** With a minimal pore diameter of ~4 Å, the predicted open state

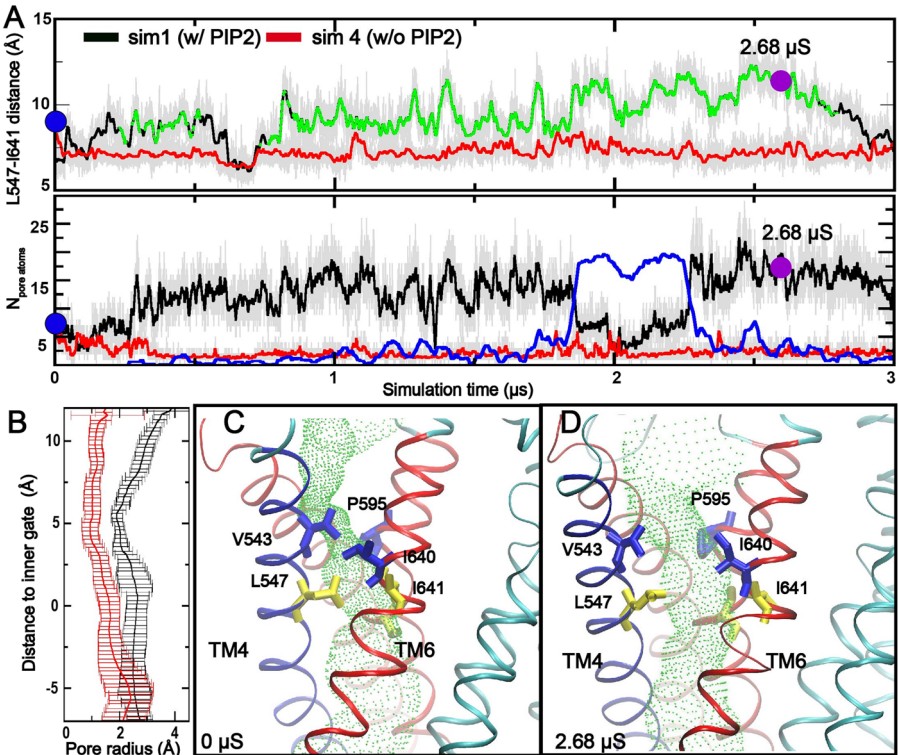

**Fig. 2 Spontaneous TMEM16A pore opening induced by specific PIP₂ binding. A** Distance between the centers of mass of L547 and I641 (upper panel) and number of water molecules (lower panel) in the neck region as a function of simulation time. The black and red traces were derived from simulations with (*sim 1, chain B*) and without (*sim 4, chain B*) PIP₂, respectively. In the upper panel, snapshots belonging to the open cluster (see Fig. S3) are highlighted in green. In the lower panel: the number of lipid heavy atoms inside the neck region during *sim 1* is also shown (blue trace). **B** Averaged pore radius profiles in the neck region calculated using HOLE[39]. The black trace was derived from snapshots sampled during 2.679–2.683 μs of *sim 1* and the red trace from those sampled during 0.0–0.01 μs of *sim 4*. **C**, **D** Representative structures of the closed and open pores, taken from 0 ns and 2681 ns of *sim 1*, respectively. TMs 3, 5, and 6 are colored in red, TM4 in blue, and the rest of the protein in cyan. Two of the inner gate residues (L547 and I641) and key pore-lining residues above the gate (V543, I640, and P595) are represented as yellow and blue sticks, respectively. The HOLE pore profiles are illustrated as the green tunnels.

of Ca²⁺-bound TMEM16A pore induced by specific PIP₂ binding should be conductive to anion permeation. Indeed, we directly observed an incidence of spontaneous diffusion of a Cl⁻ ion through the pore from the extracellular side (Fig. 3A and Supplementary Movie 4). The permeating ion largely followed the pathway identified from the HOLE[39] analysis (e.g., comparing Figs. 2D and 4A), entering through the extracellular vestibule, passing through the neck region and then exiting via the intracellular vestibule. Metadynamics simulations were performed to probe the possible permeation pathways, starting from randomly selected Cl⁻ in the intracellular vestibule. Note that, during these metadynamics simulations, Gaussian biasing potentials were accumulated along the membrane normal (*z*-axis) and the ion was free to explore any probable pathways in the membrane lateral directions (*x*- and *y*-axis). A total 17 permeation events were observed in a total of 800 ns of sampling, all of which followed similar pathways to the one observed in the unbiased simulation or predicted by the HOLE calculation (Fig. 4B). Taken together, these simulations support that the dilated pore is indeed capable of supporting Cl⁻ permeation.

During permeation, the Cl⁻ ion remain well hydrated (Fig. 3B and S6A). The ion largely maintains its solvation shell inside extracellular and intracellular vestibules, with an average of ~6 hydration waters compared to ~7 in the bulk (Fig. S6A). The hydration water number remains between 4 and 5 even in the narrowest and most hydrophobic neck region. We note this observation is different from the mechanism of Cl⁻ permeation though CLC channels, in which Cl⁻ is mostly coordinated by

protein sidechains and retains on average 1–2 hydration waters[40]. This difference in hydration level is apparently consistent with largely nonselective nature of TMEM16A towards anions and the hydrophobic nature of the narrowest neck region of the pathway. Importantly, Cl⁻ ions were observed to pause near two groups of charged residues below (K588 and K645) and above (R515, R531, R535, K603, R621, E623, E633) the largely hydrophobic neck region (Fig. S3). The apparent positive nature of the pore near the neck region is consistent with the anion selectivity of the channel, which is also consistent with previous experimental data supporting the roles of K588[8], K645[36], R515, K603, and R617[35] in anion permeation and selectivity. Several of the pore-lining basic residues have also been shown to be important for binding of the TMEM16A pore blocker NTTP[35]. The narrowest portion of the TMEM16A pore is near the inner gate and lined with residues that are predominantly hydrophobic (Fig. S7). The lack of structural features that could coordinate with permeating Cl⁻ ions in this region is not surprising, given the fact that TMEM16A is largely a nonselective anion channel.

Umbrella sampling was performed to calculate a well-converged potential of mean force (PMF) of Cl⁻ permeation through the predicted open state of TMEM16A pore. The result, shown in Fig. 4C (black trace), reveals multiple free energy barriers for Cl⁻ permeation. The largest one is ~6 kcal/mol and locates immediately above the inner gate, where the pathway is lined by three hydrophobic residues, V543, I640, and P595 (see Fig. 2D). The dilated inner gate (L547, S592, and I641) gives rise to only minor barriers of ~4 kcal/mol, likely due to the

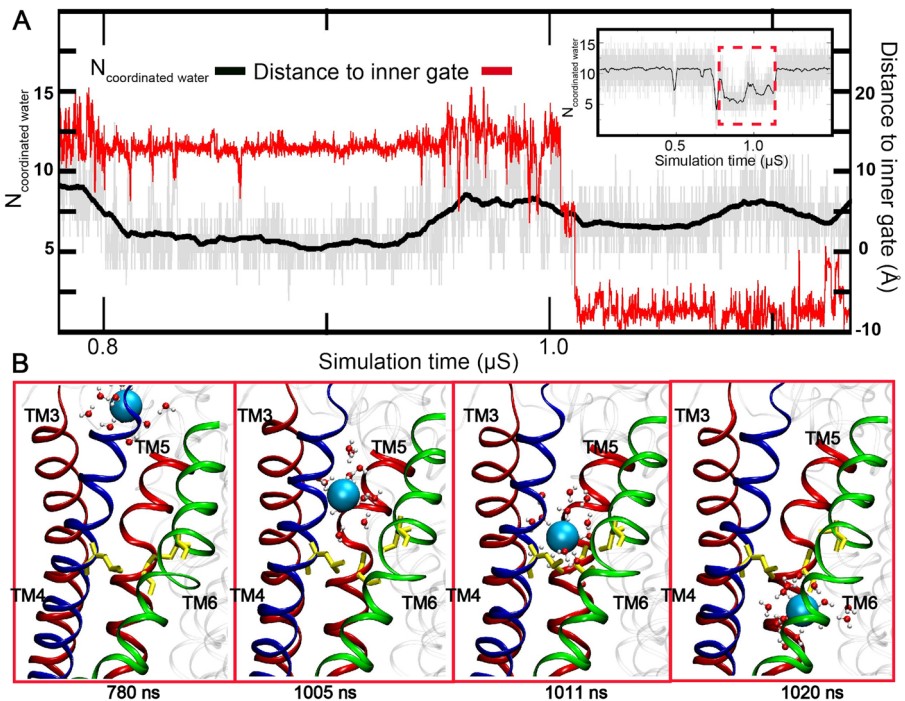

**Fig. 3 Spontaneous permeation of Cl⁻ through the TMEM16A pore. A** Number of coordinating waters of permeating Cl⁻ and its distance to the inner gate during the event (780–1030 ns of *sim 3*). The insert shows the hydrating water number of the same Cl⁻ ion in the whole simulation with the spontaneous permeation event highlighted by a dashed red box. **B** Representative snapshots during permeation. TMs 3 and 5 are represented as red cartoons and TMs 4 and 6 are colored in blue and green, respectively. The inner gate residues (L547, S592, and I641) are shown in yellow sticks. The permeating Cl⁻ ion is represented as a cyan sphere, with water molecules within 4 Å shown in the ball-and-stick mode.

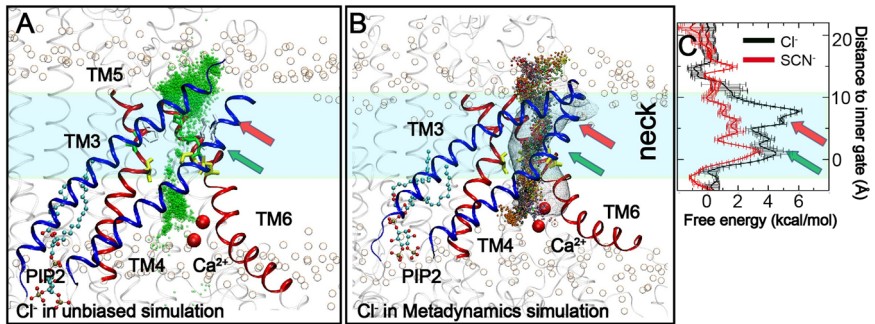

**Fig. 4 Pathway and free energy profiles of Cl⁻ permeation through TMEM16A CaCC. A** Positions of Cl⁻ (green beads) during a spontaneous permeation event (0.73–1.14 μs, *sim 3*), overlaid onto the snapshot of the channel sampled at 1.0 μs of the same trajectory. **B** Positions of permeating Cl⁻ during metadynamics simulations, overlaid onto the initial structure of metadynamics simulations. The Cl⁻ ion is presented as beads and different colors represent different permeation events. TMs 3–4 are represented as blue cartoons and TMs 5–6 as red cartoons. The rest of the protein is presented as transparent cartoons. The phosphate atoms in lipid head groups are represented as transparent spheres. The pathway predicted by HOLE is shown as a transparent mesh for reference in panel **B**. **C** Free energy profiles of Cl⁻ (black trace) and SCN⁻ (red traces) permeation through the predicted open state of TMEM16A CaCC calculated from umbrella sampling. The error bars were estimated as the difference between results calculated using the first and second halves of the sampling.

snorkeling of conserved basic residues (K588 and K645) beneath the inner gate (e.g., see Fig. S3). The presence of two adjacent layers of hydrophobic residues at the narrowest region of the pore that requires partial desolvation of Cl⁻ for passage thus accounts for all major barriers of Cl⁻ permeation. It can be expected that removing of one or both basic residues (K588 and K645) could significantly decrease the maximum conductance of the channel and/or increase the activation barrier. Conversely, replacing the ring of hydrophobic residues above the inner gate (V543, I640, and P595) with either polar or charged residues may have similar effects as inner gate residue mutations in modulating the activation of TMEM16A CaCC.

We have further estimated the maximal conductance of the predicted open state of TMEM16A CaCC by integrating the PMF profile (see Methods), which yielded a value of ~1.3 pS, which is within the experimental range of 1–8 pS for single channel maximal Cl⁻ conductance of TMEM16A[6,8,31–33]. We note that the theoretical estimate of maximum single channel conductance does not consider larger scale conformational fluctuation within the activate state and should only be considered semi-quantitative. It is possible that the open state captured in the current simulation only reflects some early stage of a large opening. Nonetheless, the pore structure properties, permeation pathway and free energy, and conductance analysis together

suggest that the predicted open state likely captures an activated TMEM16A CaCC.

**The predicted activated state of TMEM16A supports modest SCN⁻/Cl⁻ selectivity.** To further validate the putative activated state of TMEM16A, we examined the anion selectivity of SCN⁻ over Cl⁻. It has been known that TMEM16A has a peculiar preference of larger SCN⁻ over Cl⁻, with an estimated permeability ratio of $P_{SCN}/P_{Cl} \sim 6$[34,35,41]. We performed umbrella simulations to calculate the free energy of SCN⁻ permeation through the activated TMEM16A pore. The results, summarized in Fig. 4C, reveal that the maximal free energy barrier of SCN⁻ permeation is only ~3.5 kcal/mol, near the inner gate region. Intriguingly, the ring of hydrophobic residues above the inner gate does not give rise to a significant barrier to SCN⁻ permeation, in contrast to the case of Cl⁻ permeation where a large barrier of ~6 kcal/mol exists. The estimated maximal conductance of SCN⁻ is ~37 pS, which yields an apparent permeability ratio of $P_{SCN}/P_{Cl} \sim 28$. The overestimation of $P_{SCN}/P_{Cl}$ may be due to non-polarizable force fields used in this study, which does not fully reproduce the thermodynamics of ion (de)solvation[42]. It is also possible that the discrepancy is due to the potential effects of SCN⁻ on channel gating not captured by current simulation or the existence of multiple open states of TMEM16A CaCC, while the predicted activated state may only reflect one of these states.

To examine the molecular origin of different Cl⁻ and SCN⁻ conductances, we examined the detailed pore structure in the neck region. As shown in Fig. S7, the critical neck region of the TMEM16A ion permeation pathway is an irregular shaped tunnel, in contrast to symmetric channels[43,44]. The narrowest region of this tunnel above the inner gate is lined mainly by hydrophobic residues (Fig. S7C–D) and further walled by lipid tails (Fig. S7A, cyan spheres), which can partially penetrate into the gap between TMs 4 and 6. The level of desolvation required for the permeation is similar between Cl⁻ and SCN⁻ ions. The semi-hydrophobic nature of the pore surface thus cannot fully compensate for the loss of solvation, giving rise to the free barriers for ion permeation. Indeed, mean force decomposition analysis, summarized in Fig. S6B, reveals large desolvation penalties (red traces) that are partially compensated by ion–protein interactions (black traces). Importantly, with delocalized charges in SCN⁻, the free energy cost of desolvation is smaller compared to that in Cl⁻, which is the main reason for reduced free energy barrier of SCN⁻ permeation through the TMEM16A pore. This also provides a direct explanation for the observation that alanine substitution of pore-lining basic residues such as R515, K603, R621, and R792 increases selectivity of SCN⁻ over Cl⁻ [35], because reducing the hydrophilicity of the permeation pathway is more penalizing for Cl⁻ due to its larger desolvation cost. We also note that it has been previously proposed that lipids likely contribute to the formation of the ion permeation pathways in TMEM16 family proteins[33]. Results from the current simulations clearly support this idea, showing that lipids line the ion permeation pathway even for TMEM16A with limited dilation (Fig. S7A) and that the presence of lipids contributes to ion permeation properties (Fig. S6).

**Molecular mechanism of PIP₂-induced activation of TMEM16A CaCC.** We further examine the allosteric coupling mechanism between specific binding of PIP₂ to the site at the cytosolic interface of TMs 3–5 and the neck region around the inner gate between TMs 4 and 6 that undergo dilation to reactivate Ca²⁺-bound TMEM16A CaCC (e.g., see Fig. 2B). As shown in Fig. 5A, the pore opening mainly involves a "lever-like" movement of TM4. TMs 3 and 5 also undergo small

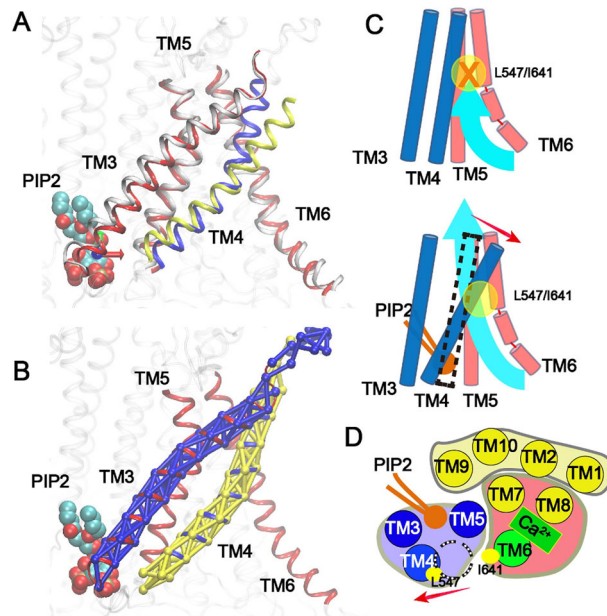

**Fig. 5 Rearrangement of TMs during PIP₂-induced TMEM16A pore reactivation. A** TM conformations in the closed (TMs 3, 5, 6: red; TM4: blue) and activated (TMs 3, 5, 6: white; TM4: yellow) states. **B** Dynamic network analysis reveals that TMs 3 and 4 are clustered into two large independent communities (blue and yellow networks, respectively). In **A** and **B**, only the headgroup of PIP₂ is shown for clarity. **C, D** Schematic illustration of the TM movement during TMEM16A pore reactivation (**C**: side view; **D**: top view). The binding of PIP₂ triggers largely rigid-body movement of TMs 3–5, particularly TM4 (highlighted by the red arrows), leading to the dilation of the pore near the neck region (yellow circles in panel **C** and dashed circle in panel **D**).

conformational transitions in response to the new TM4 configuration. In the presence of PIP₂, the PIP₂ binding residues in the C-terminal, cytosolic end of TM4 moves towards PIP₂ while the N-terminus of TM4 pivots in the reverse direction. The pivot point is maintained by the tight helix–helix packing between TMs 4 and 5 in the lower leaflet, which shows minimal movement during the opening transition. This relatively simple model of allosteric coupling is further supported by dynamic network analysis[45], which reveals that dynamic fluctuations of residues within individual TM3 and TM4 helices are highly correlated and that each TM moves as a single rigid-body-like entity (Fig. 5B–D). Dynamic pathway analysis also allows identification of so-called optimal and suboptimal paths of coupling between distal sites of the protein, which are the pathways with the strongest dynamic coupling. The result suggests that the effects of specific binding of PIP₂ to the site in the lower pore region of TMs 3–5 mainly propagate to the inner gate region through TMs 4 and 5 (Fig. S8). We have further analyzed the dynamic coupling properties of TMEM16A with and without Ca²⁺/PIP₂, and the results show that the channel has essentially the same dynamic communities and coupling pathways in both states. In particular, residues within the TM3 and 4 remain highly correlated (Fig. S9).

## Discussions

Structural studies have so far failed to provide the molecular detail of a conductive state of TMEM16A CaCC. Leveraging recent advances in GPU-accelerated atomistic simulations[46] and greatly improved protein force fields[47], as well as the recognition of a crucial role of PIP₂ in TMEM16A CaCC desensitization[24,27,30], we show that specific binding of a single

PIP$_2$ molecule to TMEM16A in the Ca$^{2+}$-bound state is sufficient to promote spontaneous opening of the pore. This allows us to determine the structural properties and anion permeation mechanism of an activated TMEM16A CaCC for the first time. The predicted open state is highly consistent with a wide range of existing mutagenesis and electrophysiological data, and the results suggest that it likely corresponds to an activated state of TMEM16A CaCC.

Besides the specific PIP$_2$ binding in the TMs 3–5 (Fig. 1), other PIP$_2$ putative binding sites have also been identified, such as near the dimer interface, intracellular loop between TMs 2, 3 and the cytoplasmic end of TM6[24,27]. Among these, only one at the dimer interface is near the specific PIP$_2$ binding[30], albeit with different sets of contacting basic residues. Furthermore, binding of PIP$_2$ to these sites has been proposed to be dynamic and form a regulatory network to modulate the channel activation. At present, it is not clear how to reconcile these important differences in the molecular basis of PIP$_2$ regulation of TMEM16A function. It has been suggested that TMEM16A can access multiple open states under different activation conditions (e.g., Ca$^{2+}$ concentration and membrane potential)[33] and that these functional states may show different responses to PIP$_2$[24,27,30]. Another important difference between the current study and Yu et al.[24] is that the previous simulations identified TM6 as the key helix that moved in response to PIP$_2$ binding. This is most likely due to distinct PIP$_2$ binding configurations investigated. The specific binding site investigated in this work locates in the back side of TM 3–5 from the pore. It is probably not a total surprise that TM 4 and 5 are observed to the main helices that respond to PIP$_2$ binding (Fig. S5A). Nonetheless, it is possible that binding of one or multiple PIP$_2$ molecules in additional sites may also contribute to activation and/or stabilization of the activated states of TMEM16A CaCC induced by specific PIP$_2$ binding. For example, the opening transition observed in the current simulations with a single PIP$_2$ bound in the TMs 3–5 site mainly involves a lever-like movement of TM4 (Fig. 5) and is transient and reversible. Binding of additional PIP$_2$ molecules to additional sites could further stabilize the open state, such as by modulating the orientation of TM6 that lines the pore opposite to TM4[24,27]. Furthermore, it should be noted that simultaneous opening of both subunits was not observed in the current simulations. While this likely reflects the stochastic nature of the process within limited simulation timeframes, it could also be because that the activated state induced by single PIP$_2$ is less stable than those induced by multiple PIP$_2$.

Both PIP$_2$ and Ca$^{2+}$ are required for sustained activation of TMEM16A CaCC[25]. Simulations of TMEM16A in the Ca$^{2+}$-free state with PIP$_2$ docked to the specific binding pocket suggest that PIP$_2$ can maintain stable contact with the same set of basic residues, R455, R486, K571, and R579, with contact probabilities of 0.24, 0.50, 0.96, and 0.99, respectively. However, the pore remained collapsed in all three simulations (Fig. S2), as expected. Analysis of the pore conformation reveals additional hydrophobic interactions between TMs 4 and 6 above the inner gate in the Ca$^{2+}$-free state (Fig. S10), which apparently help to stabilize the collapsed state of the pore. These additional contacts are absent in the simulation of the Ca$^{2+}$-bound TMEM16A with PIP$_2$. An apparent interpretation is that the straightening and hinge motion of TM 6 upon Ca$^{2+}$ binding likely strains the cross-pore contacts between TMs 4 and 6 in the upper pore region, albeit not enough to break these contacts and actually open the pore. Indeed, the cryo-EM structures also show that TMs 4 and 6 in the neck region are slightly further separated in the Ca$^{2+}$-bound state[16,24]. In addition, there are more lipid tail contacts with the upper pore hydrophobic residues (Fig. S10B), which may also help stabilize the open conformation. The Ca$^{2+}$-binding induced

strain on TMs 4 and 6 packing in the upper pore region thus primes the pore for full activation, by PIP$_2$ binding at one or multiple sites and or by other cellular stimuli such as membrane potential. Such an activation mechanism provides a basis for understanding the interplay of multiple activation and regulatory signals that control the TMEM16A function.

One of the most intriguing features of the TMEM16 family proteins is that its members can transport ions and/or phospholipid, two seemingly distinct classes of substrates, despite highly conserved structures[3,5]. Our previous study of TMEM16F lipid scramblase has suggested a conserved "clam-shell" model of activation for the TMEM16 family proteins[19]. Whether a TMEM16 protein has channel and/or scramblase activities is likely determined by how wide the same pore may be dilated (e.g., see Fig. S1), as governed by the balance of interactions at the pore and various allosteric regulatory sites. For example, while the activated state of the wild-type TMEM16A with both Ca$^{2+}$ and PIP$_2$ bound can only accommodate ion permeation, a single inner gate L543K mutation could convert TMEM16A CaCC into a lipid scramblase[19], presumably by forcing additional dilation of the pore. However, our current study suggests that the permeation pathways of ions and lipids through the pore do not fully overlap. For the wild-type TMEM16A CaCC, the ion permeation pathway consists of intracellular and extracellular vestibules connected by the neck region (Fig. 1). However, the movement of lipid head groups through the TMEM16F pore follows the so-called 'credit card' model, where lipid head groups exit or enter the pore directly from the membrane from the extracellular side, without involving the extracellular vestibules[19,38,48]. It is not clear if the ion permeation pathway will coincide with or diverge from the lipid pathway when the pore is dilated enough for scramblase activities[38]. A possible way to test this is to introduce appropriate mutations to the ring of hydrophobic residues above the inner gate (V543, P595, and I640), which is responsible for giving rise to the maximum free energy barrier of Cl$^-$ permeation. If a fully dilated scramblase shows similar conductance and ion selectivity with these mutations, it may suggest that ions mainly follow the lipid pathway. Nonetheless, the existence of alternative potential ion pathways may offer fascinating possibilities of how TMEM16 family ion channels and lipid scramblases may be regulated in biological processes.

## Methods

**Atomistic simulations.** The cryo-EM structures of mouse TMEM16A in the Ca$^{2+}$-bound (PDB: 5oyb) or Ca$^{2+}$-free (PDB: 5oyg) states[16] were used in all simulations reported in this work. The missing short loops in cytosolic domain (T260-M266, L467-F487, and L669-K682) were rebuilt using the ProMod3 tool with Swiss-PDB server[49]. The missing N- and C-terminal segments (M1-P116 and E911-L960) as well as a long loop in the cytosolic domain (Y131-V164) are presumably dynamic and thus not included in the current simulations. Residues before and after the missing segments are capped with either an acetyl group (for N-terminus) or a N-methyl amide (for C-terminus). Standard protonation states under neutral pH were assigned for all titratable residues.

As summarized in Table S1, three sets of atomistic simulations were performed. The first two sets involved the Ca$^{2+}$-bound state of TMEM16A with PO-PIP$_2$ (*sim 1-3*) and without PIP$_2$ (*sim 4-6*). The initial binding pose of PIP$_2$ (Fig. 1B) was identified using Autodock Vina[50] guided by alanine scanning data and then refined using atomistic simulations in explicit solvent and membrane, as reported previously[30]. Specifically, there is one PIP$_2$ per subunit, directly coordinated by R455, R486, K571, and R579. The third set of simulations (*sim 7-9*, 1 μs) involved the Ca$^{2+}$-free state of TMEM16A but with PIP$_2$ bound in the same specific binding site (Fig. 1B). As addition controls, we also simulated the effects of two negatively charged lipids, POPS (sim 10-12) and PI(4)P (sim 13-15), on TMEM16A. In both sets, a single POPS or PI(4)P molecule was superimposed onto the location of the PIP$_2$ in each subunit. In all simulations, no external electric field has been applied, as membrane potential is not required to activate TMEM16A under saturating Ca$^{2+}$[41]. All initial TMEM16A structures in various states were first inserted in model POPC lipid bilayers and then solvated in TIP3P water using the CHARMM-GUI web server[51]. All systems were neutralized and 150 mM KCl was added. The final simulation boxes contain about ~600 lipid molecules (POPC and/or PIP$_2$) and ~70,000 water molecules and other solutes, with a total of ~316,000 atoms and dimensions of ~150 × 150 × 135 Å$^3$. The CHARMM36m all-atom force field[47]

and the CHARMM36 lipid force field[52] were used. The PIP$_2$ parameters were adopted from a previous study[53]. All simulations were performed using CUDA-enabled versions of Amber14[54] or Gromacs 2018[55,56]. Electrostatic interactions were described by using the Particle Mesh Ewald (PME) algorithm[57] with a cutoff of 12 Å. Van der Waals interactions were cutoff at 12 Å with a smooth switching function starting at 10 Å. Covalent bonds to hydrogen atoms were constrained by the SHAKE algorithm[58], and the MD time step was set at 2 fs. The temperature was maintained at 298 K using the Nose–Hoover thermostat[59,60] (in Gromacs) or Langevin dynamics with a friction coefficient of 1 ps$^{-1}$ (in Amber). The pressure was maintained semi-isotropically at 1 bar at membrane lateral directions using the Parrinello–Rahman barostat algorithm[61] (in Gromacs) or Monte Carlo (MC) barostat method[62,63] (in Amber).

To minimize the effects of missing loop residues on the cytosolic domain, the backbone of structured region of the cytosolic domain (E121-E129, L165-R219, K228-L231, S243-T257, G267-L283, D452-S466, P890-R910) was harmonically restrained with a force constant of 1.0 kcal/(mol.Å2) during all simulations. We note that voltage-dependent gating of TMEM16A has been proposed to involve the intracellular TM2-3 linker and possibly TM6[35,41]. Nonetheless, the current simulations aim to capture voltage-independent PIP$_2$-induced activation under saturation Ca$^{2+}$. Therefore, the restraints on the structured region of the cytosolic domain is not expected to affect the activation transitions. All systems were first minimized for 5000 steps using the steepest descent algorithm, followed by a series of equilibration steps where the positions of heavy atoms of the protein/lipid were harmonically restrained[51]. Specifically, 6 equilibration steps (25 ps for steps 1–3, 100 ps for steps 4–5, and 10 ns for step 6) were performed, where the restrained force constant for proteins was set to 10, 5, 2.5, 1.0, 0.5, and 0.1 kcal mol$^{-1}$ Å$^{-2}$, respectively. For lipids, the phosphorus is restrained with force constants of 2.5, 2.5, 1.0, and 0.5, 0.1, and 0.0 kcal. mol$^{-1}$.Å$^{-2}$, respectively. In the last equilibration step, only protein heavy atoms were harmonically restrained and the system was equilibrated 10 ns in under NPT (constant particle number, pressure, and temperature) conditions. All production simulations were performed under NPT conditions at 298 K and 1 bar.

**Free energy calculations.** Well-tempered metadynamics simulations[64] and umbrella sampling[65] were used to sample ion permeation pathways and calculate the potentials of mean force (PMFs) of the permeation of Cl$^-$ and SCN$^-$ in the predicted open state of TMEM16A CaCC. For metadynamics, the initial structures were taken from representative snapshots of the unbiased simulation of Ca$^{2+}$ and PIP$_2$-bound TMEM16A (Table S1, sim 1) that locate near the center of the open-state cluster (Fig. S4, cluster 4) and have a Cl$^-$ ion in the intracellular vestibule. Only structured region of the cytosolic domain was restrained during metadynamic simulations (same as unbiased simulations). The z (membrane normal) coordinate of the selected Cl$^-$ ion inside the intracellular vestibule was used as the collective variable (CV) in metadynamics simulations. The height of the Gaussians was set to 0.01 kcal/mol with a deposition time $\tau$ of 2 ps and bias factor of 15.

For umbrella sampling, an initial structure of the open state of TMEM16A CaCC was taken from the first metadynamics simulation, at around 120 ns when the selected Cl$^-$ ion was near the inner gate (L547, S592, and I641). For umbrella sampling of SCN$^-$ permeation, the Cl$^-$ near the inner gate at the initial conformation was replaced with SCN$^-$, followed by energy minimization. The initial conformations for umbrella sampling were then generated by 10-ns steered MD simulations, during which the selected ion (Cl$^-$ or SCN$^-$) was pulled in either directions along the z-axis with the reference point moving at a velocity of 5 Å/ns. No restraint was applied to the orientation of SCN$^-$. Umbrella sampling windows were placed at 2 Å intervals and covered −5 to 24 Å with respect to the backbone COM of the inner gate. Harmonic umbrella potentials with a force constant of 0.48 kcal mol$^{-1}$ Å$^{-2}$ was applied on the distance along the membrane normal between the ion and backbone COM of three inner gate residues. To prevent occasional spontaneous closing of the pore during umbrella sampling, weak harmonic restrains with a force constant of 0.24 and 1.90 kcal/mol Å2 were applied to the backbone of TMs 3–6 and TMs 1, 2, 7–10, respectively. Each window was simulated for 20 ns, and the final PMFs were calculated using the weighted histogram analysis method (WHAM)[66]. Uncertainty of the PMF was estimated as standard errors between free-energy profiles from the first and second half of the sampling. Free energy decomposition was performed to examine the contributions from various components α (e.g., protein, lipids, and water), $W_\alpha$, to the total free energy:[67]

$$W_\alpha = -\int_{-z}^{z} dz' <F_\alpha(z')>$$

The force $F_\alpha$ between the Cl$^-$ and selected component is calculated from trajectories of umbrella sampling simulations, saved at 5 ps intervals.

**Analysis.** Unless stated otherwise, snapshots were extracted every 50 ps after 200 ns of all equilibrium MD trajectories for calculation of statistical distributions. Molecular illustrations were prepared using VMD[68]. The pore profile of the putative ion permeation pathway was calculated using program HOLE[39]. Note that the sidechains of two basic residues (K588 and K645) below the inner gate are highly mobile and can occupy the dilated pore and coordinate with permeating anions (Fig. S3). Including them in the HOLE analysis underestimates the accessible space for anion permeation in the neck region. The results shown in Fig. 2B–D thus were calculated without

including the side chains of K588 and K645. The number of water molecules inside the pore is analyzed using MDAnalysis[69] together with in-house scripts. The neck region refers to portion of the permeation path approximately from two of the inner gate residues (L547 and I641) to T518.

Clustering analysis was applied to analyze conformational states of the pore sampled during unbiased MD simulations. The features in clustering analysis included inter-helix residue–residue distances and the number of pore waters. Only pore facing residues in the narrowest neck region were considered (TM3: Y514; TM4: A42, V5343 N546, L547; TM5: S592, Y593, T594, P595; TM6: Q637, I640, I641), to maximize the sensitivity of clustering in detecting conformational states of the pore itself. All features were normalized to have close to zero mean and unit variance using a Standard Scaler method[70,71]. The time-lagged independent component analysis (tICA) method is used to reduce the dimensionality[72–74]. The simulation trajectories are clustered into micro-states using a hybrid k-centers k-medoids algorithm[71]. The generalized matrix Rayleigh quotient (GMRQ) method[74,75] is applied to optimize the hyperparameters as described in previous studies[76,77].

Single-channel conductance was estimated from the PMF G(z) as:[78,79]

$$g_{mas} = \frac{e^2}{kTL^2}\left(L^{-1}\int_{P_1} dzD(z)^{-1}e^{G(z)/kT}\right)^{-1} * \left(L^{-1}\int_{P_1} dzD(z)^{-1}e^{-G(z)/kT}\right)^{-1},$$

where k is the Boltzmann constant and T is the temperature. The integration was carried out over the pore region $P_1$ (with a length of L) where only one ion occupies the pore (−5 to 22 Å in the PMF profile; see Fig. 4). The diffusion coefficients, D(z), in the pore were estimated to be half of the bulk value[78,80]. Experimental values $D_{bulk}(Cl^-) = 203$ Å$^2$ ns$^{-1}$ and $D_{bulk}(SCN^-) = 175$ Å$^2$ ns$^{-1}$ [81] were used in the conductance estimate.

Rearrangement of TMs was examined using the community network analysis using the Network View[45]. Each node of the network represents a protein residue. The headgroup of PIP$_2$ is also considered as a single node. If two nodes form a contact (identified with a minimal heavy atom distance cutoff of 5 Å) for greater than 75% of the simulation time, an edge connecting the two nodes is added to the network. The resulting contact matrix was weighted based on the covariance of dynamic fluctuation ($C_{ij}$) calculated from MD simulations as $w_{ij} = -\log(|C_{ij}|)$. The Girvan–Newman algorithm[82] was then used to divide the network map into disjoint subnetworks (communities). This analysis allows one to identify regions of the proteins that are strongly coupled internally and the most probable pathways of how these dynamic domains are coupled to each other. The length of a path $D_{ij}$ between distant nodes i and j is defined as the sum of the edge weights between consecutive nodes along this path. The optimal pathway is identified as the shortest path determined using the Floyd–Warshall algorithm[83]. Suboptimal paths between i and j are identified as additional top paths ranked using the path length.

**Statistics and reproducibility.** The data plotted in Figs. 2, 3, and 4 is reported in Supplementary data.

The data plotted in Fig. 1B is the mean and standard from 40 snapshots extracted from trajectories (see Fig. 1 caption). The PMF plotted in Fig. 4C are derived from 18 windows, each of which contains 400,000 data. The convergence of PMF was examined by comparing results calculated using the first and second halves of the sampling, the difference of which was shown as error bars.

**Reporting summary.** Further information on research design is available in the Nature Research Reporting Summary linked to this article.

## Data availability
All relevant data are available from the authors upon request.

## Code availability
All scripts are available from the authors upon request.

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

## Acknowledgements
The authors thank Dr. Huanghe Yang for initial discussions that led to the conception of this study and for many helpful inputs throughout the study and critical reading of the manuscript. All simulations were performed on the pikes GPU cluster housed in the Massachusetts Green High-Performance Computing Cluster (MGHPCC). This work was supported by National Institutes of Health Grants R01 HL142301 (J. C.).

## Author contributions
Z. J. and J. C., conception and design of the study; Z. J., performing the simulation and analysis; Z. J. and J. C., analysis and interpretation of data, drafting, and revising the manuscript.

## Competing interests
The authors declare no competing interests.
