## [Peer Review File · Communications Biology]

Reviewers' comments:

Reviewer #1 (Remarks to the Author):

TMEM16A is a physiologically important channel that mediates chloride movement in response to Ca^{2+} signaling. Structures of TMEM16A resolved to date have provided information on the functional features of the protein, including the Ca^{2+} binding sites and the ion permeation pathways. However, none of the structures (in either Ca^{2+} -bound or Ca^{2+} -free states) has captured an open (conductive) pore conformation/state that allows Cl^- ions to go through. In this manuscript, the authors report a sizable set of atomistic MD simulations to investigate the regulating role of Ca^{2+} and the signaling lipid PIP2 in TMEM16A channel gating. Their simulations successfully capture spontaneous opening of the ion permeation pathway in the Ca^{2+} -bound state only when a PIP2 lipid is bound at a specific site. Pore opening was not observed when either Ca^{2+} or PIP2 was absent. Free energy calculations of ion conduction, for both Cl^- and SCN^- , through the open pore conformation have also been done using umbrella sampling to quantitatively describe the openness of the channel, although spontaneous permeation of one Cl^- ion was already observed during their unbiased, equilibrium simulations. The study provides insight into the activation mechanism of TMEM16A by the synergistic regulation of PIP2 and Ca^{2+} binding. This is significant, as it highlights yet another example of functional importance of lipids in regulating membrane-associated proteins.

While capturing the conformational transition of the channel by the simulations is interesting and it provides information on the nature of the open state, there are several questions that need to be addressed before I can recommend the paper for publication.

Most importantly, the mechanism provided at the end of the paper for PIP2 activation comes at a bit of surprise and needs to be elaborated with some data supporting the idea. We don't see any data on the nature of TM4 motion until the end where it is proposed to be the main change resulting in pore opening in a semi-schematic figure. The nature and extent of TM4 movement need to be established to be related to lipid binding (similar to pore radius and hydration of the pore) by data. If limited by space, the network analysis can be pushed to the SI, as it does not really make a strong point in the current form; perhaps it is not well explained.

The manuscript needs a more extensive comparison of the results with previous studies on some of core specific aspects. In particular the paper by Yu et al. (PNAS 2019) on PIP2-TMEM16A interaction is very close in scope to the present study, and some of the results of the two studies need to be compared more explicitly.

Related to the point above, Yu et al. identified three major PIP2 binding sites, claiming that these three sites constitute a network to dynamically and synergistically regulate the channel gating. Mutation of key amino acids in any of the three sites were reported by them to abolish the effect of PIP2. Although the PIP2 binding site in this study is close to one of the three sites identified in Yu et al., the results of the present study suggest that a single PIP2 can induce sufficient conformational changes to activate the channel. This is certainly an important mechanistic aspect that needs to be addressed/discussed.

Another aspect is the report of motion of TM6 in the previous study as the main effect of PIP2 binding, while TM4 is proposed to be the main element responding to PIP2 in the present study. In fact, TM6 seems to be closer to the binding site of PIP2 in the present study, and TM4 is rather far from it, although it is stated that there might be an allosteric network connecting them (not convincing yet). In light of these differences, a thorough examination of RMSDs for all helices is in order, to identify better the main elements responding to PIP2. Mechanistically relevant data can stay in the main text

while others can be delegated to the SI.

Was there a single PIP2 per subunit in the simulations? Or only one PIP2? This has not been stated explicitly in the manuscript. Based on the previous paper by the authors, it seems that they have two copies of PIP2, but that is only my guess, and needs to be clarified. Related to this, why is it "intriguing" that only one subunit shows opening in each simulation? Given that the opening is rather fast and starts in the very beginning of the simulation, it is actually worrisome as to why not all subunits show opening in the presence of PIP2 bound.

If I understand correctly, TMEM16A channel activation (as well as the PIP2 role) is voltage-dependent, with the TM2-TM3 linker and TM6 potentially involved in voltage sensing (Xiao et al, PNAS 2011; Peters et al., Neuron 2018). However, the simulations here starting from a nonconducting conformation were performed without any membrane potential. Moreover, the whole cytoplasmic domain (including the TM2-TM3 linker), which is not only dynamic and flexible but also functionally important to the channel opening, was restrained during all the simulations. This is an important problem and deserves some discussion in the manuscript.

A clear definition of the "open" conformation needs to be provided. In Fig. 2 and 2S, where the L547-I641 distance and the number of pore water are used to measure the openness of the pore, the main criterion defining open and closed pores needs to be spelled out. For example, compared to the snapshots colored as open pores, some snapshots with larger distance or more pore water are colored as closed pore (Fig. 2A).

If I understand correctly, during the free energy simulations, the channel is kept in its open state. If that is the case, there is a problem with the calculated conductance value, as it probably represents only the upper value of the macroscopic conductance measured experimentally. The channel can easily continue to flicker between open and closed microscopic states in experiments, and the measured conductance would be a weighted average of these microstates. That is not what we have in the simulations. On the other hand, the opening observed in the simulations here can represent only the beginning of a larger opening with a larger conductance. These all can affect the measured conductance and how it compares with experimental data and should be briefly mentioned.

I also note that there is a large assumption that D of the ion is half of its bulk value in the calculation of conductance. As such, perhaps we should refrain from calling the captured state a "fully activated" state.

The clusters in Fig. S4 are not clearly defined and separated. Especially, the yellow cluster, which is defined as the open and conductive state, is overlapping with the other three clusters. Please clarify the measure for the separation of the clusters.

Spontaneous diffusion of a Cl^- ion through the pore was captured in the equilibrium simulation, which could be used to seed the umbrella sampling simulations. Why was SMD used to pull a Cl^- ion through the pore?

When pulling SCN through the pore, what orientation of SCN was used and why? Please also provide details of the SMD simulation, such as the pulling velocities.

How does one explain the absence of a major barrier at the narrowest region (the neck) in the case of SCN permeation? This is really curious.

What are the protonation states of the amino acids in the Ca^{2+} -bound (with or without PIP2) and Ca^{2+} -free systems? Was there any difference between the two systems? How was it determined?

In the network analysis, the coupling of PIP2 binding to two residues (Y514 on TM3 and V543 on TM4) other than the inner gate residues (L547 on TM4 and I641 on TM6) are calculated. Please explain why these two residues were selected for the calculation rather than the inner gate residues. How can we relate the data from this calculation to the allosteric coupling mechanism between PIP2 binding and the inner gate dilation? Moreover, control analysis on the Ca²⁺-bound PIP2-free system and the Ca²⁺-free PIP2-free system should be provided to illustrate the effect of PIP2 binding in the communication between the PIP2 binding site and the gating site.

Minor Points:

Please provide snapshot pdb files of the equilibrated open pore configurations, so the reader can examine the open state.

In Fig 1, the three amino acids (two of them are mentioned in the figure legend) shown in yellow sticks in the neck region should be labelled. In Figs. 1 and 5, the coloring scheme of the key TMs in the molecular structure and cartan representation are not consistent.

In Fig 2, scale the pore radius profile in panel B to match the molecular structures in panels C and D for better illustration.

Use one unit for energy (kcal/mole or kJ/mole) and distance (nm or Å). The present presentation is confusing.

μS is the unit for conductance and should be replaced with μs (time) in Fig. 2 labels.

I suggest that most movies (if not all) be best smoothed so things can be seen better. Water can be shown as O only, if they look strange after smoothing.

In Reference 75, the first and last names of the authors are transposed. Fix.

Fig. 1 caption: Carton -> Cartoon

Fig. 1 caption: TMs 6-9 -> TMs 6-8

All-atomic -> all-atom (or atomistic)

Fig. 3: stick-and-ball --- Ball and Stick is the more common term.

Cl ions was -> Cl ions were

site in in -> site in

TMs 4 and 6 ... is slightly separated -> ... are slightly separated

It's members -> its members

Methods: Fig. S ???

Reviewer #3 (Remarks to the Author):

This manuscript explores the mechanisms of PIP2 regulation of the TMEM16A chloride channel using computational methods. TMEM16 has been shown to have a number of crucial physiological functions and the mechanisms of its regulation are poorly understood, so these studies are relevant and address an important physiological/biophysical problem. Overall, the approach yields some intriguing conclusions and insights and seems to be consistent with other published data. However, there are several major issues that dampen my enthusiasm considerably.

(1) This study has no controls. If the authors wish to make the argument that PIP2 binding to this site in the presence of Ca²⁺ stabilizes an open conformation of the channel, they need to show that other phospholipids do not produce the same effect. They should at least test PI(4)P and phosphatidylserine. Further, in their Nature Communications paper, they show that the effect of PIP2 is absent in the K567A mutant. They should show that the open conformation does not occur with this mutant and other mutants studied in this paper.

(2) This manuscript is entirely computational. Although I appreciate the power of molecular dynamic simulations, my philosophy is that computational solutions should be viewed as a predictive tool and they do not stand alone without experimental verification. I see two solutions to this problem. Preferably the authors should perform experimental tests of the model in Figure 5 or at least propose in the discussion a specific experiment that would show that this model is testable and not a pipedream. Alternatively, the authors could explore predictions made in the Le et al. paper (for example predictions about the K579-E564 salt bridge).

(3) While the authors are clearly experts in computational biology, their understanding of the TMEM16 field, as revealed by their literature citations, is rather one-sided and limited. Citations are not well chosen and are incomplete. Several examples follow, but the authors should carefully revise their manuscript to make their citation of the literature less biased and more scholarly. Reference 1 is dated 2002 and has <2 pages devoted to Ca-activated Cl channels. A better choice to cover the older literature would be: Hartzell et al. *Annu. Rev. Physiol.* 2005. 67:719–58. Reference 2 is dated 2009, but the same lab has a beautiful comprehensive review in 2014: *Physiol Rev* 94: 419–459, 2014. Reference 13 should be accompanied by Jeng et al. *J Gen Physiol.* 2016 11; 148(5):393-404 that was published in the same issue on the same topic. The section on the single channel conductance of TMEM16A overlooks reference 6 that shows a conductance of 8 pS. Incidentally, there is a nice critique of single channel data with more references in Whitlock et al. *Pflugers Arch.* 2016; 468: 455–473. On page 7 and 14, the authors cite reference 17 for showing that the L543K mutation confers lipid scramblase activity to TMEM16A. Ref 36 should also be cited because it is the first paper to show the ability to convert TMEM16A into a scramblase by point mutations in the same region.

(4) In Fig. 2A, there is surprisingly little correspondence between the N(pore water) trace and the

L547-I641 distance. For example, the channel appears to be in a closed conformation between 0.6-0.7 us while pore water remains high. Conversely, between 1.8 - 2.3 us, pore water is low, but the channel is in the open conformation. While I understand that these two measures might be expected to be temporally separated to some degree, if one is going to argue that pore waters are an indication that the channel is open, it seems necessary to show that these two are correlated. Further, what is the correspondence between PIP2 coordination by K567, R451, R575, and R482 and channel open conformation? Do the transient channel closures correspond to PIP2 unbinding? Finally, if the open pore can contain lipid instead of water, the authors should discuss this finding in relationship to the proposal by Whitlock et al. *Pflugers Arch.* 2016; 468: 455-473.

(5) I don't doubt that PIP2 can bind to this site, but I am concerned that binding in the MD simulations is simply caused by non-specific electrostatic attraction. Simulations that were performed in the Le et al. paper showed that PIP2 would spontaneously bind to its binding site within 50ns when PIP2 was placed "near" the binding pocket (the example in Fig. 4b shows this distance as <10Å). However, no binding events were observed when PIP2 was placed further away. Positive charge density of the putative binding site will attract PIP2 electrostatically, especially if PIP2 is not initially complexed with counterions like K⁺, Mg²⁺ (typically 1 -3 mM), and Ca²⁺ (probably >100 uM under conditions required to activate the channel). These divalent cations will compete with protein binding to PIP2. I would like the authors to try a less biased approach to test that this site is "the" binding site, especially because, as the authors acknowledge, other investigators have reported somewhat divergent results.

In addition to these major concerns, there are a number of specific (although not less serious) issues that require attention.

(1) On page 9, the authors conclude that Cl ions remain well hydrated during permeation and then they imply that CLC channels are similar ("This is similar to what has been observed for Cl-permeation through CLC channels, in which the number of hydrated water also drops to <5."). In fact, in CLC channels, Cl is almost completely dehydrated during permeation and virtually all of the coordination is provided by protein. If Cl permeates TMEM16 partially hydrated, this suggests that the selectivity mechanisms of ion permeation of CLC and TMEM16A are completely different. Perhaps more important is the question: Is there any experimental data supporting the idea that Cl permeates TMEM16A channels partially hydrated?

(2) Abstract: The statement that "we show that specific binding of PIP2 to TMEM16A can lead to spontaneous opening...." is not precise because it suggests that this was determined by experimental, not computational means. The abstract should be rewritten to include methodology.

(3) Page 2, the statement that the lower half of TM6 occludes the lower pore and blocks the entry of permeating ions is incorrect. While it is probably true that the lower half of TM6 unfolds during pore opening, there is no direct evidence that I know that supports the statement the authors make.

(4) Figure 1. Helices are not labelled in C. Color coding of helices in D is inconsistent with other panels.

(5) Figure 2. The y-axis is labelled N(pore water) but the blue line is lipid. Methods state that the results were determined without considering the side chains of K588 and K645. I presume this statement applies only to B-D, but the authors should be more precise and show as supplementary data the calculations with these side chains. Also, the legend states that panel A plots the "distance between the centers of mass of L547 and I641". I used centerofmass in PYMOL to calculate the center of mass of these two amino acids in 5OYB and it shows the distance is 7.3Å, not 2-3 Å as plotted here.

Exactly what was measured?

(6) Insufficient methodological detail is provided. ProMod3 requires a template. What templates were used? What information is used to determine that the models are reasonable? Also, why is water only clustered around the protein in the movies and not present in the extracellular space?

(7) The simulations were all performed with POPC bilayers, which does not mimic mammalian plasma membrane that has a significant fraction of POPS.

(8) It is stated on p7 that "The state representing the opened pore conformation is only found in simulations with PIP2", but Figure S4B shows a significant number of red dots (simulations without PIP2) in the yellow area. Further, the criteria used to define the yellow area as an "OPEN" conformation in Methods is vague. Please specify what inter-residue distances and number of waters were used.

Author Note: The original comments from the reviewers are quoted in bold fonts. Key changes to the manuscripts are noted throughout the responses.

Reviewer #1 (Remarks to the Author):

TMEM16A is a physiologically important channel that mediates chloride movement in response to Ca²⁺ signaling. Structures of TMEM16A resolved to date have provided information on the functional features of the protein, including the Ca²⁺ binding sites and the ion permeation pathways. However, none of the structures (in either Ca²⁺-bound or Ca²⁺-free states) has captured an open (conductive) pore conformation/state that allows Cl⁻ ions to go through. In this manuscript, the authors report a sizable set of atomistic MD simulations to investigate the regulating role of Ca²⁺ and the signaling lipid PIP2 in TMEM16A channel gating. Their simulations successfully capture spontaneous opening of the ion permeation pathway in the Ca²⁺-bound state only when a PIP2 lipid is bound at a specific site. Pore opening was not observed when either Ca²⁺ or PIP2 was absent. Free energy calculations of ion conduction, for both Cl⁻ and SCN⁻, through the open pore conformation have also been done using umbrella sampling to quantitatively describe the openness of the channel, although spontaneous permeation of one Cl⁻ ion was already observed during their unbiased, equilibrium simulations. The study provides insight into the activation mechanism of TMEM16A by the synergistic regulation of PIP2 and Ca²⁺ binding. This is significant, as it highlights yet another example of functional importance of lipids in regulating membrane-associated proteins.

While capturing the conformational transition of the channel by the simulations is interesting and it provides information on the nature of the open state, there are several questions that need to be addressed before I can recommend the paper for publication.

I Most importantly, the mechanism provided at the end of the paper for PIP2 activation comes at a bit of surprise and needs to be elaborated with some data supporting the idea. We don't see any data on the nature of TM4 motion until the end where it is proposed to be the main change resulting in pore opening in a semi-schematic figure. The nature and extent of TM4 movement need to be established to be related to lipid binding (similar to pore radius and hydration of the pore) by data. If limited by space, the network analysis can be pushed to the SI, as it does not really make a strong point in the current form; perhaps it is not well explained.

Responses: We agree with the reviewer's suggestion and have performed additional analysis of the movements and distributions of all TM helices in the closed and predicted open states. The result is summarized in a **new Fig S5**; it shows that TM4 is the only helix that undergoes significant movement during PIP2-induced activation.

We have also included a **short discussion to the revised manuscript on P7**, stating "The movements of TMs can be further visualized by comparing the distributions of their centers of

mass (CMs) in the closed and open states (Fig. S5A). The result show that there are ~4 and ~1 Å movements of the upper pore segments of TM4 (T539:L547) and TM3 (G510:A523) during activation, respectively, while the other TMs show minimal movements. Note that the structures of all TM helices are very stable as reflected in the small root-mean-squared fluctuation profiles (Fig. S5B).”

2 *The manuscript needs a more extensive comparison of the results with previous studies on some of core specific aspects. In particular the paper by Yu et al. (PNAS 2019) on PIP2-TMEM16A interaction is very close in scope to the present study, and some of the results of the two studies need to be compared more explicitly.*

Responses: Please see the response to Question 3 below.

3 *Related to the point above, Yu et al. identified three major PIP2 binding sites, claiming that these three sites constitute a network to dynamically and synergistically regulate the channel gating. Mutation of key amino acids in any of the three sites were reported by them to abolish the effect of PIP2. Although the PIP2 binding site in this study is close to one of the three sites identified in Yu et al., the results of the present study suggest that a single PIP2 can induce sufficient conformational changes to activate the channel. This is certainly an important mechanistic aspect that needs to be addressed/discussed.*

Responses: We agree with the reviewer (and reviewer 3; see below) that there are important unanswered questions regarding the molecular basis of PIP2 activation of TMEM16A. There is strong experimental evidence to support the role of the specific binding site investigated in the current work (Le et al, Nature Communications 2019), as well as multiple other PIP2 binding sites that potentially form a dynamic network to regulate the gating of TMEM16A (Yu et al PNAS 2019). These observations likely reflect the complexity of TMEM16A regulation and existence of multiple activated states accessible under different experimental conditions (*e.g.*, Ca²⁺ concentration). Reconciling these results requires additional experiments (and simulations) beyond the scope of this study. Instead, motivated by the identification of the specific binding site, the objective of this computational study is to test if binding a single specific PIP2 would be sufficient to activate the channel and if so what are the structure features of the predicted activated state.

We fully agree with the reviewer that these complexities should be further discussed, particularly in the context of results reported in Yu et al PNAS 2019. We have followed the suggestion and included an extended discussion of these remaining issues and how our observations are consistent and different from those reported in Yu et al 2019 study (see **P14-15**):

“Besides the specific PIP2 binding in the TMs 3-5 (Fig. 1), other PIP2 putative binding sites have also been identified, such as near the dimer interface, intracellular loop between TMs 2, 3 and the

cytoplasmic end of TM6.^{25, 28} Among these, only one at the dimer interface is near the specific PIP2 binding³¹, albeit with different sets of contacting basic residues. Furthermore, binding of PIP2 to these sites has been proposed to be dynamic and form a regulatory network to modulate the channel activation. At present, it is not clear how to reconcile these important differences in the molecular basis of PIP2 regulation of TMEM16A function. It has been suggested that TMEM16A can access multiple open states under different activation conditions (e.g., Ca^{2+} concentration and membrane potential)³⁴ and that these functional states may show different responses to PIP2.^{25, 28, 31} Another important difference between the current study and Yu et al²⁵ is that the previous simulations identified TM6 as the key helix that moved in response to PIP2 binding. This is most likely due to distinct PIP2 binding configurations investigated. The specific binding site investigated in this work locates in the back side of TM3-5 from the pore. It is probably not a total surprise that TM4 and 5 are observed to be the main helices that respond to PIP2 binding (Fig. S5A).....”

4 Another aspect is the report of motion of TM6 in the previous study as the main effect of PIP2 binding, while TM4 is proposed to be the main element responding to PIP2 in the present study. In fact, TM6 seems to be closer to the binding site of PIP2 in the present study, and TM4 is rather far from it, although it is stated that there might be an allosteric network connecting them (not convincing yet). In light of these differences, a thorough examination of RMSDs for all helices is in order, to identify better the main elements responding to PIP2. Mechanistically relevant data can stay in the main text while others can be delegated to the SI.

Was there a single PIP2 per subunit in the simulations? Or only one PIP2? This has not been stated explicitly in the manuscript. Based on the previous paper by the authors, it seems that they have two copies of PIP2, but that is only my guess, and needs to be clarified. Related to this, why is it “intriguing” that only one subunit shows opening in each simulation? Given that the opening is rather fast and starts in the very beginning of the simulation, it is actually worrisome as to why not all subunits show opening in the presence of PIP2 bound.

Responses: We would like to clarify that the specific binding site actually locates at TM3-5 behind the pore and is far from TM6 (Fig 1). PIP₂ directly interacts with TM3/4 but not TM6.

We have slightly revised the manuscript (see **P17**) to clarify that: “Specifically, there is one PIP₂ per subunit, directly coordinated by R455, R486, K571 and R579.”

Previous experimental study suggests the two subunit is activated independently (**P2 and ref 14, 15**). We suspect that it is coincidental that we did not observe both subunits opening in our three simulations, due to the limited timescale and transient nature of the opening transition.

5 *If I understand correctly, TMEM16A channel activation (as well as the PIP2 role) is voltage-dependent, with the TM2-TM3 linker and TM6 potentially involved in voltage sensing (Xiao et al, PNAS 2011; Peters et al., Neuron 2018). However, the simulations here starting from a nonconducting conformation were performed without any membrane potential. Moreover, the whole cytoplasmic domain (including the TM2-TM3 linker), which is not only dynamic and flexible but also functionally important to the channel opening, was restrained during all the simulations. This is an important problem and deserves some discussion in the manuscript.*

Responses: We thank the reviewer for pointing out the potential pitfalls of restraining the cytosolic domains implied in voltage gating. The simulations with PIP2 (or other lipids bound) were initiated with the Ca²⁺ bound structure and thus mimicked the condition under saturating Ca²⁺ where membrane depolarization is not required for activation. Therefore, we believe that the weak restraints on the structured portion of the cytosolic domain, imposed to minimize the effects of missing loops, should not have a significant impact on observing PIP2-induced activation.

The following notes have been made in Methods: “*In all simulations, no external electric field has been applied, as membrane potential is not required to fully activate TMEM16A under saturating Ca²⁺.*” (on **P17**), and later “*We note that voltage-dependent gating of TMEM16A has been proposed to involve the intracellular TM2-3 linker and possibly TM6.^{36, 43} Nonetheless, the current simulations aim to capture voltage-independent PIP₂-induced activation under saturation Ca²⁺. Therefore, the restraints on the structured region of the cytosolic domain is not expected to affect the activation transitions.*” (on **P18**)

6 *A clear definition of the “open” conformation needs to be provided. In Fig. 2 and 2S, where the L547-I641 distance and the number of pore water are used to measure the openness of the pore, the main criterion defining open and closed pores needs to be spelled out. For example, compared to the snapshots colored as open pores, some snapshots with larger distance or more pore water are colored as closed pore (Fig. 2A).*

Responses: We have added to following discussion to the revised manuscript to clarify how the open and close states were identified (on **P7**): “*As the pore opening involves dilating in the whole upper pore region, a single residue-residue distance or the number of pore water molecules alone could not clearly sperate the open and closed state. Here, we performed cluster analysis based on distances between pore-lining residues and the number of pore waters in the neck region (See Method part for details).*” The pathway properties of the resulting clusters were then inspected to assign each cluster to open, closed or other (transient) states.

7 *If I understand correctly, during the free energy simulations, the channel is kept in its open state. If that is the case, there is a problem with the calculated conductance value, as it probably represents only the upper value of the macroscopic conductance measured experimentally. The*

channel can easily continue to flicker between open and closed microscopic states in experiments, and the measured conductance would be a weighted average of these microstates. That is not what we have in the simulations. On the other hand, the opening observed in the simulations here can represent only the beginning of a larger opening with a larger conductance. These all can affect the measured conductance and how it compares with experimental data and should be briefly mentioned.

Responses: We generally agree with the reviewer's points but want to point out that the maximum single channel conductance measurements reported in the literature should have been corrected for less than 100% open probabilities. We have revised the manuscript to further clarify the pitfalls in comparing the calculated conductance and experimental measurements (see **P10-11**), "*We note that the theoretical estimate of maximum single channel conductance does not consider larger scale conformational fluctuation within the activate state and should only be considered semi-quantitative. It is possible that the open state captured in the current simulation only reflects some early stage of a large opening.*"

8 *I also note that there is a large assumption that D of the ion is half of its bulk value in the calculation of conductance. As such, perhaps we should refrain from calling the captured state a "fully activated" state.*

Responses: We agree and have removed the wording "fully" from "fully activated" throughout the revised manuscript.

9 *The clusters in Fig. S4 are not clearly defined and separated. Especially, the yellow cluster, which is defined as the open and conductive state, is overlapping with the other three clusters. Please clarify the measure for the separation of the clusters.*

Responses: We have added the following clarification to the Figure S4 caption: "*Note that the clustering was performed on multi-dimensional tICA and some clusters may appear to substantially overlap in the 2D projection.*"

10 *Spontaneous diffusion of a Cl⁻ ion through the pore was captured in the equilibrium simulation, which could be used to seed the umbrella sampling simulations. Why was SMD used to pull a Cl⁻ ion through the pore?*

Responses: The pore underwent significant conformational fluctuations during the spontaneous ion permeation. For the free energy calculation, we want to calculate the free energy profile for the representative conformational state selected. Therefore, SMD was used to generate the initial states of umbrella sampling.

11 *When pulling SCN through the pore, what orientation of SCN was used and why? Please also provide details of the SMD simulation, such as the pulling velocities.*

Responses: There is no restraint imposed on the orientation and SCN is allowed to freely tumble. The following statement has been added (see **P19**), “... during which the selected ion (Cl⁻ or SCN⁻) was pulled in either directions along the z-axis with the reference point moving at a velocity of 5Å/ns. No restraint was applied to the orientation of SCN.”

12 *How does one explain the absence of a major barrier at the narrowest region (the neck) in the case of SCN permeation? This is really curious.*

Responses: There is still a major barrier for SCN⁻ permeation near the neck region, albeit lower than that of Cl⁻ (3.5 vs 4.5 kcal/mol, see green arrow in Fig. 4C). Please also note that Cl⁻ experiences a second and higher barrier of ~6 kcal/mol above the inner gate (red arrow in Fig. 4C). We have further performed free energy decomposition analysis to understand the physical basis of lower barrier for SCN⁻ permeation (Fig. S6) and the analysis is discussed on **P12, 2nd paragraph**.

13 *What are the protonation states of the amino acids in the Ca²⁺-bound (with or without PIP2) and Ca²⁺-free systems? Was there any difference between the two systems? How was it determined?*

Responses: We assigned the standard protonation states for all titratable residues expected at neutral pH, including those of the carboxyl groups coordinating Ca²⁺. We note that it has been suggested that Ca²⁺ coordinating residue E623 remains deprotonated in both Ca²⁺-bound and free states. Protonation of this residue with acidic condition can activate Ca²⁺-free TMEM16A. (Silvia Cruz-Rangel, J Physiol. 2017).

The following statement has been added at **P17**: “Standard protonation states under neutral pH were assigned for all titratable residues.”

14 *In the network analysis, the coupling of PIP2 binding to two residues (Y514 on TM3 and V543 on TM4) other than the inner gate residues (L547 on TM4 and I641 on TM6) are calculated. Please explain why these two residues were selected for the calculation rather than the inner gate residues. How can we relate the data from this calculation to the allosteric coupling mechanism between PIP2 binding and the inner gate dilation? Moreover, control analysis on the Ca²⁺-bound PIP2-free system and the Ca²⁺-free PIP2-free system should be provided to illustrate the effect of PIP2 binding in the communication between the PIP2 binding site and the gating site.*

Responses: Y514 and V543 were selected here because they are where the more prominent barrier of Cl⁻ permeation locates (above the inner gate, Fig. 4C red arrow). We have performed additional analysis of the dynamic coupling properties in both Ca²⁺ bound and free states; the results are summarized in a new SI figure (Fig.S9), showing essentially the same dynamic communities and coupling pathways as in Fig. 5B. We have included a short statement in the revised manuscript (see P14): “We have further analyzed the dynamic coupling properties of TMEM16A with and without Ca²⁺/PIP2, and the results show that the channel has essentially the same dynamic communities and coupling pathways in both states. In particular, residues within the TM3 and 4 remain highly correlated (Fig.S9).”

Minor Points:

Please provide snapshot pdb files of the equilibrated open pore configurations, so the reader can examine the open state.

Responses: We have added a new SI file (SI_sim12_2681ns_B.pdb), which is the pdb structure of TMEM16A in the open state illustrated in Fig 2.

“SI_sim12_2681ns_B.pdb”

In Fig 1, the three amino acids (two of them are mentioned in the figure legend) shown in yellow sticks in the neck region should be labelled. In Figs. 1 and 5, the coloring scheme of the key TMs in the molecular structure and carton representation are not consistent.

In Fig 2, scale the pore radius profile in panel B to match the molecular structures in panels C and D for better illustration.

Use one unit for energy (kcal/mole or kJ/mole) and distance (nm or Å). The present presentation is confusing.

μS is the unit for conductance and should be replaced with μs (time) in Fig. 2 labels.

Responses: All of above have been corrected as suggested.

I suggest that most movies (if not all) be best smoothed so things can be seen better. Water can be shown as O only, if they look strange after smoothing.

Responses: The movies were compressed to reduce the file size and thus display slightly reduced clarity. We show the water hydrogen to better illustrate how it coordinates with ions and/or protein sidechains.

In Reference 75, the first and last names of the authors are transposed. Fix.

Fig. 1 caption: Carton -> Cartoon

Fig. 1 caption: TMs 6-9 -> TMs 6-8

All-atomic -> all-atom (or atomistic)

Fig. 3: stick-and-ball --- Ball and Stick is the more common term.

Cl ions was -> Cl ions were

site in in -> site in

TMs 4 and 6 ... is slightly separated -> ... are slightly separated

It's members -> its members

Methods: Fig. S ???

Responses: Thank you for the careful reading. These typos and errors have been corrected.

Author Note: The original comments from the reviewers are quoted in bold fonts. Key changes to the manuscripts are noted throughout the responses.

Reviewer #3 (Remarks to the Author):

This manuscript explores the mechanisms of PIP2 regulation of the TMEM16A chloride channel using computational methods. TMEM16 has been shown to have a number of crucial physiological functions and the mechanisms of its regulation are poorly understood, so these studies are relevant and address an important physiological/biophysical problem. Overall, the approach yields some intriguing conclusions and insights and seems to be consistent with other published data. However, there are several major issues that dampen my enthusiasm considerably.

(1) This study has no controls. If the authors wish to make the argument that PIP2 binding to this site in the presence of Ca²⁺ stabilizes an open conformation of the channel, they need to show that other phospholipids do not produce the same effect. They should at least test PI(4)P and phosphatidylserine. Further, in their Nature Communications paper, they show that the effect of PIP2 is absent in the K567A mutant. They should show that the open conformation does not occur with this mutant and other mutants studied in this paper.

Responses: We appreciate the reviewer's suggestion of additional controls to show that other lipids binding to the same site would not induce pore opening. We first want to note that the same site is always occupied by POPC in existing control simulations without PIP2, which do not induce pore opening. We followed the reviewer's suggestion and perform 6 additional control simulations with PI(4)P and POPS lipids docked to the specific binding site using the same pose as in PIP2 simulations (*sim 10-15, SI Table S1*). Setup of these new control simulations is described in the revised Methods section (**P17**). The results are summarized in a **revised SI Fig. S2** and discussed in **P7**:

*"It should be noted the binding pockets are always occupied with POPC lipids even in simulations without PIP2. These observations are consistent with the experimental results showing that PIP2 is required for maintaining the conductive state of TMEM16A even under saturating Ca²⁺ concentrations.^{26, 27, 31} As additional controls, we further examined the effects of two negatively charged lipids, POPS and PI(4)P, bound to the same pockets. Previous experimental studies have suggested that POPS could not activate TMEM16A, while PI(4)P only show minor effects on inhibiting channel rundown.^{26, 31} As summarized in Fig. S2, the binding of these negatively charged lipids only slightly increased the numbers of pore water (*sim 10-15*) compared to those without PIP2 (and with POPC) (*sim 7-9*). Interestingly, an transient pore-opening event was observed in one of the simulations with PI(4)P (*sim 13*), apparently consistent with some capacity of PI(4)P in re-activating the channel."*

For K567A mutant, it likely abolishes PIP2 specific binding and thus activation. Our current simulations do not attempt to make de novo predictions on what kinds of lipids can bind to the binding site or how pocket mutations may perturb the binding properties. Such prediction requires binding free energy analysis, which is beyond the scope of this work and extremely challenging due to the size and flexibility of lipid molecules. As also explained in our response to Reviewer #1, Question #3 above, our current study was motivated by the identification of the specific binding site and the objective is to test if binding a single specific PIP2 would be sufficient to activate the channel and if so what are the structure features of the predicted activated state.

(2) This manuscript is entirely computational. Although I appreciate the power of molecular dynamic simulations, my philosophy is that computational solutions should be viewed as a predictive tool and they do not stand alone without experimental verification. I see two solutions to this problem. Preferably the authors should perform experimental tests of the model in Figure 5 or at least propose in the discussion a specific experiment that would show that this model is testable and not a pipedream. Alternatively, the authors could explore predictions made in the Le et al. paper (for example predictions about the K579-E564 salt bridge).

Responses: We wholeheartedly agree with the notion that computation must be integrated with experiment to derive reliable understanding of complex channel function. We want to note that our simulations are motivated by and build on our previous experimental and computational studies (with Huanghe Yang's lab). Throughout this study, we compare key structural and functional properties to published experimental data (mutagenesis, conductance, selectivity etc).

We have followed the reviewer's advice and propose a set of specific experiments that could be used to test key predictions from this computational study (see **P10**): *“It can be expected that removing of one or both basic residues (K588 and K645) could significantly decrease the maximum conductance of the channel and/or increase the activation barrier. Conversely, replacing the ring of hydrophobic residues above the inner gate (V543, I640 and P595) with either polar or charged residues may have similar effects as inner gate residue mutations in modulating the activation of TMEM16A CaCC.”*

We further hypothesize that (see **P16-17**): *“A possible way to test this is to introduce appropriate mutations to the ring of hydrophobic residues above the inner gate (V543, P595 and I640), which is responsible for giving rise to the maximum free energy barrier of Cl⁻ permeation. If a fully dilated scramblase shows similar conductance and ion selectivity with these mutations, it may suggest that ions mainly follow the lipid pathway.”*

(3) While the authors are clearly experts in computational biology, their understanding of the TMEM16 field, as revealed by their literature citations, is rather one-sided and limited. Citations are not well chosen and are incomplete. Several examples follow, but the authors should

carefully revise their manuscript to make their citation of the literature less biased and more scholarly. Reference 1 is dated 2002 and has <2 pages devoted to Ca-activated Cl channels. A better choice to cover the older literature would be: Hartzell et al. Annu. Rev. Physiol. 2005. 67:719–58. Reference 2 is dated 2009, but the same lab has a beautiful comprehensive review in 2014: Physiol Rev 94: 419–459, 2014. Reference 13 should be accompanied by Jeng et al. J Gen Physiol. 2016 11; 148(5):393-404 that was published in the same issue on the same topic. The section on the single channel conductance of TMEM16A overlooks reference 6 that shows a conductance of 8 pS. Incidentally, there is a nice critique of single channel data with more references in Whitlock et al. Pflugers Arch. 2016; 468: 455–473. On page 7 and 14, the authors cite reference 17 for showing that the L543K mutation confers lipid scramblase activity to TMEM16A. Ref 36 should also be cited because it is the first paper to show the ability to convert TMEM16A into a scramblase by point mutations in the same region.

Responses: We sincerely appreciate the reviewer's efforts to bring our attention to these key publications. We have updated the reference and incorporate them into the discussions throughout the revised manuscript. Specifically,

- New reference added as 4, 5, 15. Ref 36 (now 39) has been added to when discuss the mutations:
- **P8:** “*can confer lipid scramblase activity to TMEM16A, presumably by promoting wider dilation of the same ion conducting pore.*^{19, 39}”
- **P16:** “*without involving the extracellular vestibules.*^{19, 39, 50} *It is not clear if the ion permeation pathway will coincide with or diverge from the lipid pathway when the pore is dilated enough for scramblase activities.*³⁹”
- **P4:** “*single channel maximal Cl conductance in the range of below 1 and up to 8 pS.*^{6, 8, 32, 33, 34} *suggesting that the predicted open state may correspond to an activated TMEM16A CaCC that has evaded structural studies so far.*”
- **P10:** “*which is within the experimental range of 1-3 8 pS for single channel maximal Cl-conductance of TMEM16A.*^{6, 8, 32, 33, 34}”

(4) *In Fig. 2A, there is surprisingly little correspondence between the N(pore water) trace and the L547-I641 distance. For example, the channel appears to be in a closed conformation between 0.6-0.7 us while pore water remains high. Conversely, between 1.8 - 2.3 us, pore water is low, but the channel is in the open conformation. While I understand that these two measures might be expected to be temporally separated to some degree, if one is going to argue that pore waters are an indication that the channel is open, it seems necessary to show that these two are correlated. Further, what is the correspondence between PIP2 coordination by K567, R451, R575, and R482 and channel open conformation? Do the transient channel closures correspond to PIP2 unbinding? Finally, if the open pore can contain lipid instead of water, the authors should discuss this finding in relationship to the proposal by Whitlock et al. Pflugers Arch. 2016; 468: 455–473.*

Responses: We agree that the number of pore water should in principle provide an intuitive indicator of the state of the channel. However, the channel can undergo relatively small conformational fluctuations and lead to transient increase in pore hydration. We observe that no single metrics (e.g., pore water, helix distance etc) can fully resolve the conformational state. As such, we performed clustering analysis to first identify major conformational states and then assigned them to open, close or other states. This provides a superior approach to resolve the open and close states. We have included a short clarification in the revised manuscript (see **P7**): *“As the pore opening involves dilating in the whole upper pore region, a single residue-residue distance or the number of pore water molecules alone could not clearly separate the open and closed state. Here, we performed cluster analysis based on distances between pore-lining residues and the number of pore waters in the neck region (See Method part for details) ...”*

The revised manuscript also includes the probabilities of PIP2 contacts with the coordinating basic residues in the binding pocket (see **P5**): *“During the simulations, PIP2 maintained stable contacts with the coordinating residues, mainly R455, R486, K571 and R579. The contact probabilities of PIP2 to these four residues are 0.40, 0.49, 0.97 and 0.99, respectively. The average RMSD of the PIP2 headgroup and these basic residues from the initial conformation is $3.5 \pm 0.7 \text{ \AA}$ ”*

We thank the reviewer for bringing to our attention of the nice discussion on how lipid tails contribute to the permeation pathway. The revised manuscript now includes an expanded discussion on this aspect (see **P12**): *“We also note that it has been previously proposed that lipids likely contribute to the formation of the ion permeation pathways in TMEM16 family proteins.³⁴ Results from the current simulations clearly support this idea, showing that lipids line the ion permeation pathway even for TMEM16A with limited dilation (Fig. S7A) and that the presence of lipids contribute to ion permeation properties (Fig. S6).”*

(5) I don't doubt that PIP2 can bind to this site, but I am concerned that binding in the MD simulations is simply caused by non-specific electrostatic attraction. Simulations that were performed in the Le et al. paper showed that PIP2 would spontaneously bind to its binding site within 50ns when PIP2 was placed “near” the binding pocket (the example in Fig. 4b shows this distance as <10Å). However, no binding events were observed when PIP2 was placed further away. Positive charge density of the putative binding site will attract PIP2 electrostatically, especially if PIP2 is not initially complexed with counterions like K⁺, Mg²⁺ (typically 1 -3 mM), and Ca²⁺ (probably >100 uM under conditions required to activate the channel). These divalent cations will compete with protein binding to PIP2. I would like the authors to try a less biased approach to test that this site is “the” binding site, especially because, as the authors acknowledge, other investigators have reported somewhat divergent results.

Responses: We appreciate the reviewer's comments on our published work (Le et al, Nature Communications 2019). As explained above in our responses to Question #1 and Reviewer #1

Question #3, the objective of this work is not to further establish if the specific PIP2 binding site reported in our previous work is true. Instead, our objective of this computational study is to test if binding a single specific PIP2 would be sufficient to activate the channel and if so what are the structure features of the predicted activated state.

Regarding the feasibility of direct simulation of spontaneous binding of PIP2 to a binding site, this is actually extremely challenging due to the size and flexibility of PIP2 and very slow diffusion of lipid molecules (especially towards a relative occluded pocket near the dimer interface under study here). The timescale of such binding process can be easily ms or longer, beyond the reach of atomistic MD in general. Advanced and/or approximate approaches are generally needed, such as HMMM used in Yu et al PNAS (2019).

In addition to these major concerns, there are a number of specific (although not less serious) issues that require attention.

(1) On page 9, the authors conclude that Cl ions remain well hydrated during permeation and then they imply that CLC channels are similar (“This is similar to what has been observed for Cl-permeation through CIC channels, in which the number of hydrated water also drops to <5.”). In fact, in CLC channels, Cl is almost completely dehydrated during permeation and virtually all of the coordination is provided by protein. If Cl permeates TMEM16 partially hydrated, this suggests that the selectivity mechanisms of ion permeation of CLC and TMEM16A are completely different. Perhaps more important is the question: Is there any experimental data supporting the idea that Cl permeates TMEM16A channels partially hydrated?

Responses: We appreciate the reviewer comment and have revised the manuscript to note (see P9): “We note this observation is different from the mechanism of Cl- permeation through CLC channels, in which Cl- is mostly coordinated by protein sidechains and retains on average 1-2 hydration waters.⁴¹ This difference in hydration level is apparently consistent with largely nonselective nature of TMEM16A towards anions and the hydrophobic nature of the narrowest neck region of the pathway.”

(2) Abstract: The statement that “we show that specific binding of PIP2 to TMEM16A can lead to spontaneous opening...” is not precise because it suggests that this was determined by experimental, not computational means. The abstract should be rewritten to include methodology.

Responses: The wording “we show” has been replaced with “.. atomistic simulations show” in the abstract.

(3) Page 2, the statement that the lower half of TM6 occludes the lower pore and blocks the entry of permeating ions is incorrect. While it is probably true that the lower half of TM6 unfolds during pore opening, there is no direct evidence that I know that supports the statement the authors make.

Responses: We agree that the notion of occlusion is not correct. TM6 placement in the calcium free state narrows the lower opening but does “occlude” the entrance. The statement has been removed in the revised manuscript (see **P2**).

(4) Figure 1. Helices are not labelled in C. Color coding of helices in D is inconsistent with other panels.

Responses: This has been corrected as suggested.

(5) Figure 2. The y-axis is labelled N(pore water) but the blue line is lipid. Methods state that the results were determined without considering the side chains of K588 and K645. I presume this statement applies only to B-D, but the authors should be more precise and show as supplementary data the calculations with these side chains. Also, the legend states that panel A plots the “distance between the centers of mass of L547 and I641”. I used centerofmass in PYMOL to calculate the center of mass of these two amino acids in 5OYB and it shows the distance is 7.3A, not 2-3 A as plotted here. Exactly what was measured?

Responses: We apologize for the mistake on the scale of the y axis of L547-I641 distance in Fig 2A. The figure has been corrected. We have noted that the same method been applied to pore analysis shown in Fig 2 B-D (see **P20**). The highly dynamic nature of K588 and K645 can lead to spuriously small pore profiles. As noted in **P5**, “The charged sidechains of K588 and K645 could rotate and point towards the inner gate region when the hydrophobic inner gate residues, L547 on TM4 and I641 on TM6, become separated during the opening transition.”

(6) Insufficient methodological detail is provided. ProMod3 requires a template. What templates were used? What information is used to determine that the models are reasonable? Also, why is water only clustered around the protein in the movies and not present in the extracellular space?

Responses: ProMod3 was only used to generate the initial structure of short loops missing in the cryo-EM structures (PDB: 5oyb and 5oyg). The resulting structures were energy minimized and thoroughly equilibrated in explicit water and membrane as described in the **Methods**.

The SI movies only water molecules near the TM 4-6 for clarity. This is now noted in the revised movie description.

(7) The simulations were all performed with POPC bilayers, which does not mimic mammalian plasma membrane that has a significant fraction of POPS.

Responses: POPC bilayers have been widely used to model biological membranes. There is not clear experimental evidence on how the presence of POPS in mammalian plasma may modulate TMEM16A CaCC activation. As such, we have opted to use the simplest model bilayers such that one can focus on the effects of specific binding of various lipids including PIP2, POPC, POPS and PI(4)P on TMEM16A activation.

(8) It is stated on p7 that “The state representing the opened pore conformation is only found in simulations with PIP2”, but Figure S4B shows a significant number of red dots (simulations without PIP2) in the yellow area. Further, the criteria used to define the yellow area as an “OPEN” conformation in Methods is vague. Please specify what inter-residue distances and number of waters were used.

Responses: As explained in response to Reviewer 1 (Question 9), “... clustering was performed on multi-dimensional tICA and some clusters may appear to substantially overlap in the 2D projection.” The properties considered in tICA clustering are detailed on **P20**, “The features used in clustering analysis included inter-helix residue-residue distances and the number of pore waters. Only pore facing residues in the narrowest neck region were considered (TM3: Y514; TM4: A42, V5343 N546, L547; TM5: S592, Y593, T594, P595; TM6: Q637, I640, I641), to maximize the sensitivity of clustering in detecting conformational states of the pore itself. All features were normalized to have close to zero mean and unit variance using a Standard Scaler method^{72, 73.}”

Author Note: The original comments from the reviewers are quoted in bold fonts. Key changes to the manuscripts are noted throughout the responses.

Reviewer #4 (Remarks to the Author):

The manuscript reports interesting insight into the binding of PIP2 to the TMEM16A channel. It was recently discovered that PIP2 will directly interact with TMEM16A, thereby affecting the function of the channel. This work shows that PIP2 binding with calcium ions can open an inner gate at the neck of the pore and enlarge the pore. This process is mainly due to TM4 movement. (The inner gate here refers to the narrowest radius of the channel-the neck area) This result is helpful to understand the mechanism of PIP2 regulating TMEM16A.

The following major and minor comments should be addressed.

Major comments:

1: PIP2 binds to the lower part of TMs 3-5. Why does it cause the upper part of TM4 to bend? Although the author provides an allosteric coupling mechanism, the realization of this process should check the effect of the outer leaflet's phospholipids on TM4 because they Direct contact with TM4. In addition, the conformation of TM4 in the system with and without calcium ions should be checked in the absence of PIP2. Because the TM4 of 5oyb and 5oyg are almost the same. Will the bind of PIP2 affect TM3 and TM5?

Responses: A new analysis has been performed to investigate the lipid contacts in the upper pore region. The results are summarized in a **new SI Fig S10B**, and discussed in the revised manuscript (see **P16**): *“In addition, there are more lipid tail contacts with the upper pore hydrophobic residues (Fig S8B), which may also help stabilize the open conformation.”*

In all simulations, TMs remains stable; the dynamics and movements of all TMs are analyzed in a **new Fig. S5** and explained below in response to Question #2.

2: A recent review summarizes the open and closed structure of the TMEM16 family. TM4 and TM6 have opposite movement trends (Shi sai et al., CSBJ, 2020). Did you find that TM6 has reverse motion in the simulation? Please provide a side view of the simulated initial state and the open state to intuitively display the conformational changes of the protein. In addition, in the absence of PIP2, does POPC occupy the binding site of PIP2?

Responses: We have performed new analysis to illustrate the movement of all TMs. The results are summarized in a **new SI Fig. S5A** and discussed in the revised manuscript (see **P17**): *“The movements of TMs can be further visualized by comparing the distributions of their centers of mass (CMs) in the closed and open states (Fig. S5A). The result show that there are ~4 and ~1 Å*

movements of the upper pore segments of TM4 (T539:L547) and TM3 (G510:A523) during activation, respectively, while the other TMs show minimal movements. Note that the structures of all TM helices are very stable as reflected in the small root-mean-squared fluctuation profiles (Fig. S5B)."

We have also provided a PDB file of open state of TMEM16A as a **new SI material** ("SI_sim12_2681ns_B.pdb"), which will be more straightforward for reader to compare against the cyro-EM structures.

POPC does occupy the binding pocket when not occupied by other lipids such as PIP2. This is clarified in the revised manuscript (**at P7**): "*It should be noted the binding pockets are always occupied with POPC lipids even in simulations without PIP2.*"

3: The topic of the article is the promotion of PIP2 on calcium-activated TMEM16A. Therefore, the details of the interaction between PIP2 and protein are very critical. However, the details of the binding are not shown in the article. I think this is important. And, whether PIP2 is stable at the binding site, please show the RMSD of PIP2 in the whole process. Another earlier study showed that PIP2 has many binding sites (Yu K, et al. PNAS, 2019, 116 (40): 201904012.) Why did the author only pay attention to this one? TM6 is more related to the gating function, why is there no research on TM6 binding with PIP2.

Responses: We agree with the reviewer's criticisms, which have also been echoed in points raised by the other two reviewers. Please refer to our responses to Reviewer #3, Question #4 (PIP2 contacts and stability) and Reviewer #1, Question #3 (alternative PIP2 binding sites) and Question #4 (TM6 movement). The stability of Pip2 binding as well as its RMSD is discussed at P5.

Minor comments:

1: page 16. Simulate why the two softwares Amber14 and Gromacs 2018 are used. Which dynamics simulations were performed by amber14 and which were performed by Gromacs. Please list the details of the simulation implementation.

Responses: This been clarified in the revised Table S1.

2: page 16. followed by a series of equilibration steps where the positions of heavy atoms of the protein/lipid were harmonically restrained with restrained force constants gradually decreased from 10 to 0.1 kcal / (mol. Å²). Please describe this process in detail.

Responses: More details have been added to Methods (see **P18**): "*Specifically, 6 equilibration step (25 ps for steps 1-3, 100 ps for steps 4-5 and 10 ns for step 6) were performed, where the restrained force constant for proteins was set to 10, 5, 2.5, 1.0, 0.5 and 0.1 kcal/(mol.Å²), respectively. For*

lipids, the phosphorus is restrained with force constants of 2.5, 2.5, 1.0 and 0.5, 0.1 and 0.0 kcal/(mol.Å²), respectively.”

3: Please mark the name of the helix and residue in Figure 3B.

Responses: Labels have been added.

REVIEWERS' COMMENTS:

Reviewer #1 (Remarks to the Author):

The manuscript has been improved during the revision and while addressing the criticism offered by the reviewers. Nevertheless, the authors have been a bit dismissive of some of the fundamental questions, perhaps most importantly regarding the lack of enough control (brought up by another reviewer), and the fact that opening is not reproducibly observed in all subunits after PIP2 binding (my review). At the very least, the shortcoming regarding reproducibility has to be clearly added to the discussion of the paper (they can use the same explanation they use in their response letter). I understand that the stochastic nature of the process might prevent them from observing the event in their short timescale, but for the same stochasticity argument, what the opening they observe in only one case can be completely irrelevant event not having anything to do with the physiological way the channel opens.

Reviewer #2 (Remarks to the Author):

My concerns have been addressed.

Reviewer #3 (Remarks to the Author):

The authors have done a very nice job revising this paper and responding to my comments. However, there remains one point that would benefit from some discussion. The authors conclude that the single channel conductance with SCN as permeant ion is significantly greater than with Cl. However, whole-cell currents in the literature with SCN are usually smaller than with Cl, despite the fact that SCN permeability is greater than Cl as determined by reversal potential measurements under bi-ionic conditions. One explanation is that SCN affects channel gating so that open probability is greatly reduced in the whole-cell experiments. There is some data available to support this suggestion. It would be nice if the authors commented on this.

Also, I noticed some grammatical errors and typos – the authors should carefully proofread the manuscript.

Line 266 should reference Fig. S5B not 5B.

Line 292 should reference Fig. S4B.

The PDB file SI_sim12_2681ns.pdb should include PIP2. It would also be good to include a snapshot with PIP2 unbound.

REVIEWERS' COMMENTS:

Reviewer #1 (Remarks to the Author):

The manuscript has been improved during the revision and while addressing the criticism offered by the reviewers. Nevertheless, the authors have been a bit dismissive of some of the fundamental questions, perhaps most importantly regarding the lack of enough control (brought up by another reviewer), and the fact that opening is not reproducibly observed in all subunits after PIP2 binding (my review). At the very least, the shortcoming regarding reproducibility has to be clearly added to the discussion of the paper (they can use the same explanation they use in their response letter). I understand that the stochastic nature of the process might prevent them from observing the event in their short timescale, but for the same stochasticity argument, what the opening they observe in only one case can be completely irrelevant event not having anything to do with the physiological way the channel opens.

Response: We have followed the suggestion of the reviewer and added the following discussions: “*This is likely due to the stochastic nature of the opening transition and a consequence of the limited simulation timescale.*” (Page 5), and “*Furthermore, it should be noted that simultaneous opening of both subunits was not observed in the current simulations. While this likely reflects the stochastic nature of the process within limited simulation timeframes, it could also be because that the activated state induced by single PIP2 is less stable than those induced by multiple PIP2.*” (Page 15)

Reviewer #2 (Remarks to the Author):

My concerns have been addressed.

Response: We greatly appreciate the support!

Reviewer #3 (Remarks to the Author):

The authors have done a very nice job revising this paper and responding to my comments. However, there remains one point that would benefit from some discussion. The authors conclude that the single channel conductance with SCN as permeant ion is significantly greater than with Cl. However, whole-cell currents in the literature with SCN are usually smaller than with Cl, despite the fact that SCN permeability is greater than Cl as determined by reversal potential measurements under bi-ionic conditions. One explanation is that SCN affects channel gating so that open probability is greatly reduced in the whole-cell experiments. There is some data available to support this suggestion. It would be nice if the authors commented on this.

Response: We appreciate the reviewer's comment about the SCN permittivity. The following discussion has been made at P12: “*The overestimation of PSCN/PCI It is also possible the discrepancy is due to to the potential effects of SCN on channel gating not captured by current simulation or to the existence of multiple open states of TMEM16A CaCC, while the predicted activated state may only reflect one of these states.*”

Also, I noticed some grammatical errors and typos – the authors should carefully proofread the manuscript.

Line 266 should reference Fig. S5B not 5B.

Line 292 should reference Fig. S4B.

The PDB file SI_sim12_2681ns.pdb should include PIP2. It would also be good to include a snapshot with PIP2 unbound.

Response: This has been corrected. The PDB file has been updated to include PIP2 and an additional PDB file is now provided in the PIP2 unbound state.